# Social and psychological adversity are associated with distinct mother and infant gut microbiome variations

Health disparities are driven by underlying social disadvantage and psychosocial stressors. However, how social disadvantage and psychosocial stressors lead to adverse health outcomes is unclear, particularly when exposure begins prenatally. Variations in the gut microbiome and circulating proinflammatory cytokines offer potential mechanistic pathways. Here, we interrogate the gut microbiome of mother-child dyads to compare high-versus-low prenatal social disadvantage, psychosocial stressors and maternal circulating cytokine cohorts (prospective case-control study design using gut microbiomes from 121 dyads profiled with 16 S rRNA sequencing and 89 dyads with shotgun metagenomic sequencing). Gut microbiome characteristics significantly predictive of social disadvantage and psychosocial stressors in the mothers and children indicate that different discriminatory taxa and related pathways are involved, including many species of *Bifidobacterium* and related pathways across several comparisons. The lowest inter-individual gut microbiome similarity was observed among high-social disadvantage/high-psychosocial stressors mothers, suggesting distinct environmental exposures driving a diverging gut microbiome assembly compared to low-social disadvantage/ low-psychosocial stressors controls ($P = 3.5 \times 10^{-5}$ for social disadvantage, $P = 2.7 \times 10^{-15}$ for psychosocial stressors). Children's gut metagenome profiles at 4 months also significantly predicted high/low maternal prenatal IL-6 ($P = 0.029$), with many bacterial species overlapping those identified by social disadvantage and psychosocial stressors. These differences, based on maternal social and psychological status during a critical developmental window early in life, offer potentially modifiable targets to mitigate health inequities.

Health inequities experienced by socially disadvantaged populations are irrefutable and urgent societal issues[1,2]. Psychosocial factors contributing to these inequities often begin early in life and include the prenatal environment, which has profound and lifelong effects on fetal and infant outcomes[3–5]. How social disadvantage (SD) and psychosocial stressors (PS) become biologically embedded and then cause disparate health outcomes remain unclear. Many SD/PS-related morbidities are associated with systemic chronic inflammation[6]. The gut microbiome (GM) can shape and modulate the immune system[7,8], and is associated with systemic inflammation and autoimmune diseases, many of which are found at higher rates in populations experiencing SD[9–11].

The GM is itself shaped by environmental factors including the consequences of hardships such as diet, housing, and stress. Despite increasing awareness of an intersection between the GM and psychosocial inequities[12,13], few studies[14–16] have examined their impact on

✉ e-mail: warnerbb@wustl.edu; mmitreva@wustl.edu

human gut microbial community structure and function, particularly in the perinatal period. GM studies in pregnant women have (i) focused on maternal psychological state[17–21] or socioeconomic status[22–24] (but not both), (ii) used 16 S rRNA analysis, which limits taxonomic and metabolic profiling of GMs, and (iii) not paired maternal and infant biologic transfer of bacteria and inflammatory phenotypes.

Understanding the dynamic interaction between the GM and social determinants of health is of interest because of the potential for alterability that may lessen potential GM-related health impacts[25,26]. This approach would be particularly promising for the perinatal period, a critical interval in which perturbations in GM community structure and function have long-lasting effects[27–29]. Using a prospective birth cohort whose mothers are enrolled during gestation, we identify the distinct association of exposure to SD and PS on GM structure and function for mothers and their infants. By classifying mothers and their infants as "high" or "low" SD and PS, we identify discriminatory taxa driving this association using whole metagenomic shotgun (WMS) sequencing. Furthermore, we investigate a potential mechanistic link between the GM and host response by simultaneously examining maternal prenatal circulating cytokines.

## Results and discussion

A set of 121 mother–child dyads was drawn prospectively from a larger parent study, the Early Life Adversity Biological Embedding and Risk for Developmental Precursors of Mental Disorders Study (eLABE)[30,31], based on eligibility criteria that included infants having reached 4 months of life, with available maternal prenatal 3rd trimester and infant 4 month feces (see Methods and Supplementary Fig. 1). The calculation of the SD and PS latent factor variables were calculated based on an a priori hypothesis (as previously published[30], and previously used to identify significant SD and PS-associated differences in brain composition at birth[32,33]). SD and PS status (see Methods for computation score) had substantial variability (see cohort characteristics in Table 1 and Supplementary Data 1a). Details of statistics for significance values in Supplementary Data 1a are provided in Supplementary Tables 1b–e.

### High-level taxonomic profiling mother:child GM diversity varies by SD/PS status, but the GM diversity is significantly associated with SD and PS only in children

To provide an overview of GM profiles, we performed targeted bacterial 16 S rRNA gene sequencing (see Methods) for the 121 mother-child dyads (Fig. 1). Across the 121 mother-child dyads (Fig. 1a), SD and PS scores were significantly but only moderately correlated ($r = 0.395$, degrees of freedom = 119, $P = 7.2 \times 10^{-6}$, effect size statistic = 4.70, 95% confidence interval = −1.98:1.98, two-sided-test; Fig. 1b), corroborating the literature[34] and demonstrating the importance of differentiating effects between these two constructs. Samples were classified as "high" and "low" SD and PS based on the score distribution across samples (Fig. 1c; ≥+0.5 and ≤−0.5 standard deviations above and below the average). Several variables were significantly associated with high-vs-low SD and PS scores (Table 1). As expected, variables that were components of SD and PS were also significantly associated, as these variables were used as inputs to calculate the composite scores[30] (Supplementary Data 1a).

Across all samples, 3072 amplicon sequence variants (ASVs) representing unique 16 S rRNA nucleotide sequences were detected (abundance data provided per sample in Supplementary Data 2, nucleotide sequences are provided in Supplementary Data 3). In mothers, SD and PS scores had negative but not significant correlations with α-diversity (within-sample, measured by Faith phylogenetic diversity[35]) based on the bacterial taxonomic composition, consistent with previous reports of decreased α-diversity associated with lower Socioeconomic Status (SES)[14–16] (Fig. 2b; Supplementary Data 1f shows complete correlation statistics). In contrast, among children,

α-diversity was significantly positively correlated with SD ($r = 0.579$, degrees of freedom = 119, $P = 3.6 \times 10^{-12}$, effect size statistic = 7.74, 95% confidence interval = −1.98:1.98, two-sided-test) and PS ($r = 0.327$, degrees of freedom = 119, $P = 2.5 \times 10^{-4}$, effect size statistic = 3.78, 95% confidence interval = −1.98:1.98, two-sided-test; Fig. 2b). This observation may be partly explained by the significantly lower frequency of breastfeeding among high-SD mothers compared to the low-SD mothers ($P = 2.1 \times 10^{-5}$, $\chi^2 = 18.10$, 95% confidence interval = −∞:3.84; Table 1, Fig. 2b) as breastfed infants have lower GM α-diversity compared to formula-fed infants[36]. Similar results were seen with α-diversity calculated using the Shannon diversity index (Supplementary Fig. 2), indicating a robust association regardless of the metric used.

An examination of dissimilarity (β-diversity, weighted UniFrac distance) between mother and child GMs showed significant negative correlations with SD ($r = -0.438$, degrees of freedom = 119, $P = 5.0 \times 10^{-7}$, effect size statistic = −5.31, 95% confidence interval = −1.98:1.98, two-sided-test) and PS ($r = -0.295$, degrees of freedom=119, $P = 1.0 \times 10^{-3}$, effect size statistic = −3.37, 95% confidence interval = −1.98:1.98, two-sided-test). This trend holds with several other measures of β-diversity including Bray–Curtis distance, Aitchison distance[37], and unweighted UniFrac[38] distance (Supplementary Fig. 3). This may be partially due to formula-fed/high-SD infants having a GM that is more adult-like compared to low-SD children, as reflected by their increased alpha diversity. To further investigate differences between the overall microbiome profiles, we performed the Dirichlet Multinomial Mixtures (DMM) clustering approach for metagenomics[39] to cluster all of the mother and child samples based on the relative abundance of the taxa in their GMs. Using this approach, we identified that the GMs of mothers overall clustered separately from children, with the samples separating optimally into two clusters that separated the children (cluster 1, 94.5% of cluster members) from the mothers (cluster 2, 99.1% of cluster members) (visualized on an NDMS plot in Fig. 3a). For all clustering, statistical support for cluster numbers using DMM log-posterior loss correction[39] (lplc), silhouette scores[40] and prediction strength[41] is provided in Supplementary Data 1g.

Using just the mother GM samples, we identified optimal clustering for two clusters (visualized on an NMDS plot in Fig. 3b), but the clusters did not significantly differ in SD or PS (Two-sided Mann–Whitney $U$-test, FDR-adjusted $P = 0.944$, n1 = 60, n2 = 61, $U = 1844$, 95% confidence interval = 1451.9:2208.1, standardized effect size=0.0064 and $P = 0.944$, n1 = 60, n2 = 61, $U = 1896$, 95% confidence interval=1451.9:2208.1, standardized effect size=0.031, respectively; Supplementary Data 1e). We also identified two sample clusters as being optimal for the children GM samples, with cluster 1 having significantly greater SD scores than cluster 2 (two-sided Mann–Whitney $U$-test; FDR-adjusted $P = 2.7 \times 10^{-9}$, n1 = 61, n2 = 60, $U = 3021$, 95% confidence interval=1451.9:2208.1, standardized effect size = 0.56), and significantly greater PS scores in cluster 1 than 2 (two-sided Mann–Whitney $U$-test $P = 6.9 \times 10^{-3}$, $n = 61$, n2 = 60, $U = 2395$, 95% confidence interval = 1451.9:2208.1 standardized effect size=0.27; Fig. 3c), suggesting that high-SD and high-PS scores are associated with a distinct overall GM profile in children, but not mothers. As shown in Fig. 3d, the high-SD / high-diversity GM children tend to have much lower frequencies of breast milk feeding, which has been previously associated with higher GM diversity[36].

We next compared β-diversity between each sample dyad within and between SD and PS groups using weighted UniFrac distance[38] (Fig. 4). High-SD mothers had significantly more variable GMs than those of low-SD mothers (two-sided, FDR-corrected Mann–Whitney $U$-test $P = 4.3 \times 10^{-6}$, n1 = 595, n2 = 903, $U = 307209$, 95% confidence interval=252585.6:284699.4, standardized effect size = 0.12; Supplementary Data 1e). This effect was even stronger for the PS comparison (two-sided FDR-corrected Mann–Whitney $U$-test $P = 2.3 \times 10^{-7}$, n1 = 630, n2 = 496, $U = 185377$, 95% confidence interval = 145622.3:166857.7, standardized effect size = 0.16), with low-PS

**Table 1 | Participant characteristics at study entry for each of the primary comparisons of interest**

| Comparison | Statistic | | All 16S samples | MGS samples | | | | |
|---|---|---|---|---|---|---|---|---|
| | | | | All | Social Disadvantage | | Psychosocial Stressors | |
| | | | | | Low | High | Low | High |
| | **# Sample pairs (Mother and Child)** | | **121** | **89** | **35** | **43** | **36** | **32** |
| Maternal | Mother delivery age (years) | Min | 18.8 | 19.3 | 25.7 | 19.3 | 21.5 | 19.3 |
| | | Max | 41.3 | 41.3 | 41.3 | 38.7 | 41.3 | 41.0 |
| | | Average ± Std. dev. | 29.8 ± 5.1 | 30.4 ± 5.2 | 33.2 ± 4.4 | 28.5 ± 5.2 | 32.0 ± 4.8 | 28.4 ± 5.4 |
| | | P-value (T-test) | | | $3.5 \times 10^{-5}$ | | $4.5 \times 10^{-3}$ | |
| | Race | African American | 52.1% | 53.9% | 2.9% | 93.0% | 36.1% | 75.0% |
| | | Caucasian | 44.6% | 42.7% | 91.4% | 7.0% | 61.1% | 21.9% |
| | | Other | 3.3% | 3.4% | 5.7% | 0.0% | 2.8% | 3.1% |
| | | P-value (Chi square test) | | | $6.1 \times 10^{-15}$ | | $1.3 \times 10^{-3}$ | |
| | Social Disadvantage Score (SD) | Min | −2.2 | −2.2 | −2.2 | 0.4 | −2.2 | −2.2 |
| | | Max | 1.3 | 1.3 | −0.8 | 1.3 | 1.3 | 1.3 |
| | | Average ± Std. dev. | −0.24 ± 1.01 | −0.23 ± 1.1 | −1.48 ± 0.39 | 0.78 ± 0.26 | −0.71 ± 1.11 | 0.31 ± 0.81 |
| | | P-value (M-W U-test) | | | $4.2 \times 10^{-14}$ | | $4.0 \times 10^{-4}$ | |
| | Psychosocial Stressors Score (PS) | Min | −1.7 | −1.7 | −1.7 | −1.4 | −1.7 | 0.34 |
| | | Max | 2.4 | 2.4 | 1.3 | 2.4 | −0.73 | 2.4 |
| | | Average ± Std. dev. | −0.21 ± 0.86 | −0.21 ± 0.98 | −0.70 ± 0.75 | 0.12 ± 0.91 | −1.13 ± 0.25 | 0.92 ± 0.51 |
| | | P-value (M-W U-test) | | | $3.9 \times 10^{-5}$ | | $1.5 \times 10^{-12}$ | |
| Children | Birthweight (g) | Min | 2200 | 2200 | 2760 | 2200 | 2300 | 2200 |
| | | Max | 4665 | 4627 | 4370 | 4627 | 4370 | 4270 |
| | | Average ± Std. dev. | 3319 ± 538 | 3283 ± 556 | 3538 ± 459 | 3077 ± 548 | 3379 ± 555 | 3042 ± 494 |
| | | P-value (T-test) | | | $1.3 \times 10^{-4}$ | | 0.010 | |
| | Gestational age (weeks) | Min | 37 | 37 | 37 | 37 | 37 | 37 |
| | | Max | 41 | 41 | 41 | 41 | 41 | 41 |
| | | Average ± Std. dev. | 39.0 ± 1.1 | 39.0 ± 1.1 | 39.5 ± 0.95 | 38.6 ± 1.03 | 39.1 ± 1.1 | 38.7 ± 1.1 |
| | | P-value (M-W U-test) | | | $7.2 \times 10^{-4}$ | | 0.24 | |
| | Child sex | Female | 43.8% | 40.4% | 42.9% | 39.5% | 41.7% | 46.9% |
| | | Male | 56.2% | 59.6% | 57.1% | 60.5% | 58.3% | 53.1% |
| | | P-value (Chi square test) | | | 0.77 | | 0.67 | |
| | Route of delivery[a] | NSVD | 65.3% | 64.0% | 60.0% | 67.4% | 69.4% | 59.4% |
| | | VAVD | 6.6% | 4.5% | 5.7% | 4.7% | 2.8% | 9.4% |
| | | Cesarean section | 28.1% | 31.5% | 34.3% | 27.9% | 27.8% | 31.3% |
| | | P-value ($\chi^2$) NSVD vs Caes | | | 0.52 | | 0.61 | |
| | | P-value ($\chi^2$) VAVD vs others | | | 0.83 | | 0.25 | |
| | Breast milk feeding frequency | ≥50% | 47.9% | 46.1% | 71.4% | 23.3% | 58.3% | 31.3% |
| | | <50% | 52.1% | 53.9% | 28.6% | 76.7% | 41.7% | 68.8% |
| | | P-value (Chi square test) | | | $2.1 \times 10^{-5}$ | | 0.025 | |

[a]NSVD Normal spontaneous vaginal delivery, VAVD vacuum assisted vaginal delivery.
"Low" and "High" SD and PS scores are separated according to the distribution of the metadata as shown in Fig. 1. More complete metadata comparisons are available in Supplementary Data 1a, normality test statistics in Supplementary B, T-test statistics are in Supplementary Data 1c, Chi-square test statistics in Supplementary Data 1d and Mann–Whitney U-test statistics in Supplementary Data 1e. All tests are two-sided when applicable.

mothers having the most similar GMs out of the comparisons (Fig. 4a). High-SD children had the most similar GM profiles (Fig. 4b), which potentially relates to their increased α-diversity (Fig. 2b). Low-SD children also have some overall similarity in their GM profiles, but the low-SD and high-SD children GMs have little similarity to each other compared to the similarity among low-SD and among high-SD children (two-sided Wilcoxon rank sum test $P < 10^{-15}$, n1 = 1505, n2 = 903, $U = 844197$, 95% confidence interval = 647134.1:711880.9, standardized effect size = 0.2 and $P < 10^{-15}$, n1 = 595, n2 = 1505, $U = 278513$, 95% confidence interval = 423196.2:472278.8, standardized effect size = 0.29, respectively), consistent with clustering results in Fig. 3c. The same is true for PS in the children, except the within-group similarity for low-PS and high-PS was not significantly

different. In addition to weighted UniFrac distance (Supplementary Fig. 4a), we also identified similar results using other metrics of β-diversity including Bray-Curtis diversity (Supplementary Fig. 4b), Aitchison distance (which showed similar trends but less significant results; Supplementary Fig. 4c) and unweighted UniFrac distance (which showed more diversity between high-PS samples in the children, and lower distance for high-SD mothers; Supplementary Fig. 4d), also indicating a robust statistical association regardless of the metric used. Overall, the greater similarity between GM of low-SD and low-PS mothers compared to high-SD and high-PS mothers suggest distinct environmental exposures that either converge or diverge maternal GM, highlighting the interface of SD/PS, environment and GM.

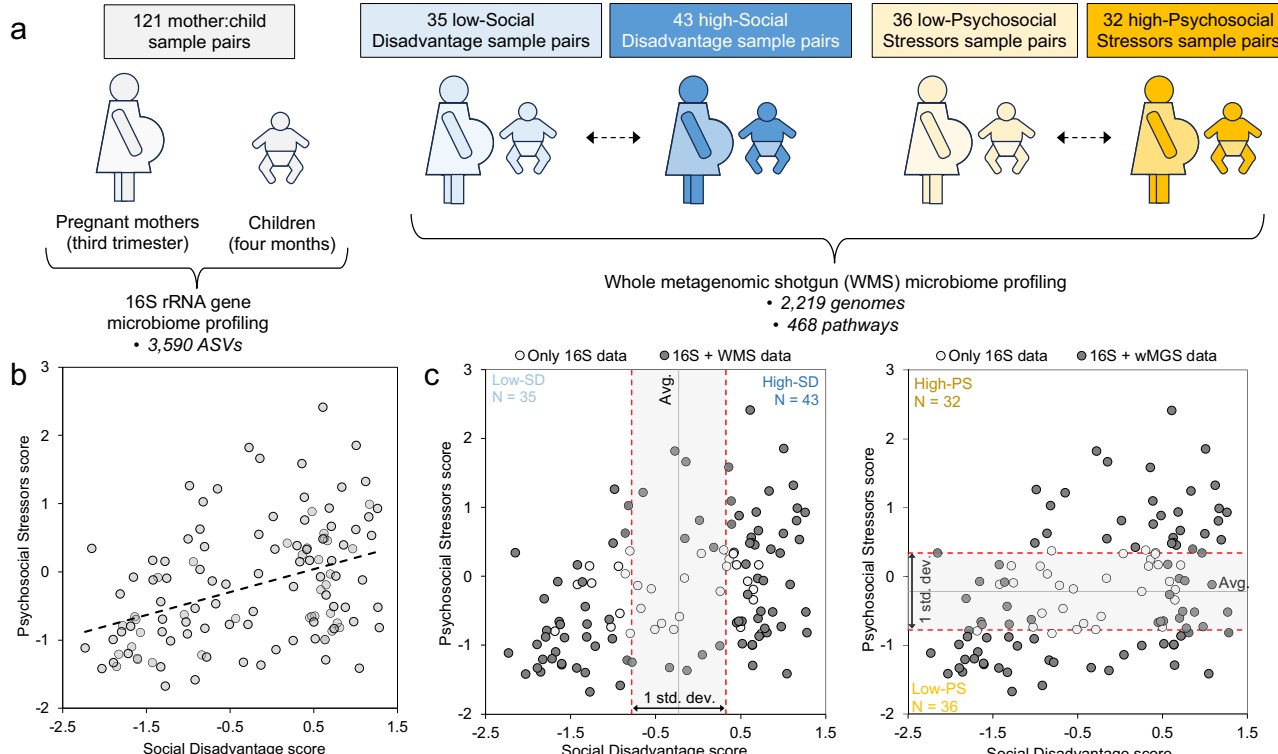

**Fig. 1 | Overview of cohorts. a** Portrayal of the 16 S rRNA and WMS (whole metagenome shotgun) sample sets. **b** Social Disadvantage (SD) scores are positively correlated with Psychosocial Stressors (PS) scores (across the 121 sample pairs). **c** The 89 WMS samples are divided into "low-SD", "high-SD", "low-PS" and "high-PS" groups based on the distribution of values (samples within 1 standard deviation of the average value for each variable are excluded).

## Species-level gut microbiome species and pathway reconstructions identify conserved and distinct features between mothers and children

We performed WMS sequencing on 89 of the 121 mother–child dyads. These pairs were selected based on the distribution of SD and PS scores so that extremes of "high" and "low" samples were compared (Fig. 1c). GM taxonomic profiles were generated from an average of 6 Gb reads/sample using the Unified Human Gastrointestinal Genome (UHGG[42]) database (Version 1), identifying 2,219 bacterial genomes with detection in any sample, 1,274 of which were detected in at least 3 maternal samples or 3 child samples (Supplementary Fig. 5a). Twelve genomes were detected frequently across all samples (≥40% of mothers and children; Supplementary Fig. 5b), including six Bifidobacteria (*B. breve*, *B. bifidum*, *B. catenulatum*, *B. pseudocatenulatum*, *B. adolescentis*, and *B. infantis*, the most frequently detected genome across all samples; 69.7% of mothers and 89.9% of children; Supplementary Fig. 6), two *Bacteroides* species (*B. dorei* and *B. xylanisolvens*), two *Faecalicatena* species (*F. gnavus* and *F. unclassified*), *Flavonifractor plautii* and *Eggerthella lenta*. Four of these species were also identified as "core mother-infant shared species" in a previous WMS study[43], but the limited overlap may be a result of different analysis approach (clade-specific marker genes from MetaPhlAn2[43,44] vs. mapping to UHGG[42]) and/or difference in sequence depth of coverage.

We next quantified metabolic pathways[45] (Supplementary Data 2) to compare the functional potential of the GM communities (using HUMAnN3[46–48], version 3). Of 468 pathways detected in any sample, 94% (438) were detected in at least 3 mother or 3 child GMs (Supplementary Fig. 5c). In contrast to the genomes with relatively sparse identification across samples, almost half (46.3%) of detected pathways were identified in ≥90% of samples in both mothers and children (top right of the plot, Supplementary Fig. 5d), including 130 pathways (27.8%) detected in all 178 samples (89 mothers and 89 children). Despite the taxonomic differences between the GM of mothers and

children (31.7% shared genomes; Supplementary Fig. 5a), 87.2% of the pathways were encoded by both mothers and children's GMs (Supplementary Fig. 5c), an observation reported previously and attributed to shared "core" microbial community functions essential for all species despite the distinct populations of species adapted to different diets at different stages of life[43].

## Bacterial species including members of Lawsonibacter and Bifidobacterium distinguish the GMs of mothers based on SD and PS

To dissociate the impact of the highly inter-related SD and PS on the maternal GM and to identify discriminatory bacterial taxa that are strongly associated with either or both scores, we analyzed taxonomic and pathway GM profiles using three statistical approaches. First, we used supervised Random Forest (RF[49]) machine-learning to (i) quantify the ability to accurately predict SD and PS classification based on the mothers' and the children's GMs based on binomial distribution tests (Fig. 5a; Supplementary Data 1h) and receiver operating characteristic (ROC) curves (Fig. 5b) (ii) identify the specific genomes and pathways that most strongly differentiate between high and low SD and PS scores (Figs. 6 and 7; Table 2; Supplementary Tables 1i and 1j; see Methods). Second, as a validation of RF results, we used linear discriminant analysis effect size (LEfSe[50]) to test differential genome and pathway abundance by calculating Kruskal–Wallis *P*-values and linear discriminant analysis (LDA) effect sizes. Third, we performed differential abundance testing with ANCOM-BC2[51] as an orthogonal approach to provide additional confidence in specific results.

We identified a set of SD- and PS-discriminatory bacteria where relative abundances classified mothers into low-SD or high-SD with 70.5% accuracy ($P = 2.5 \times 10^{-3}$, FDR-corrected binomial distribution test, Supplementary Data 1h; AUC = 0.794, $P = 2.0 \times 10^{-4}$, two-sided Wilcoxon rank sum test calculated using the "roc.area" function in the

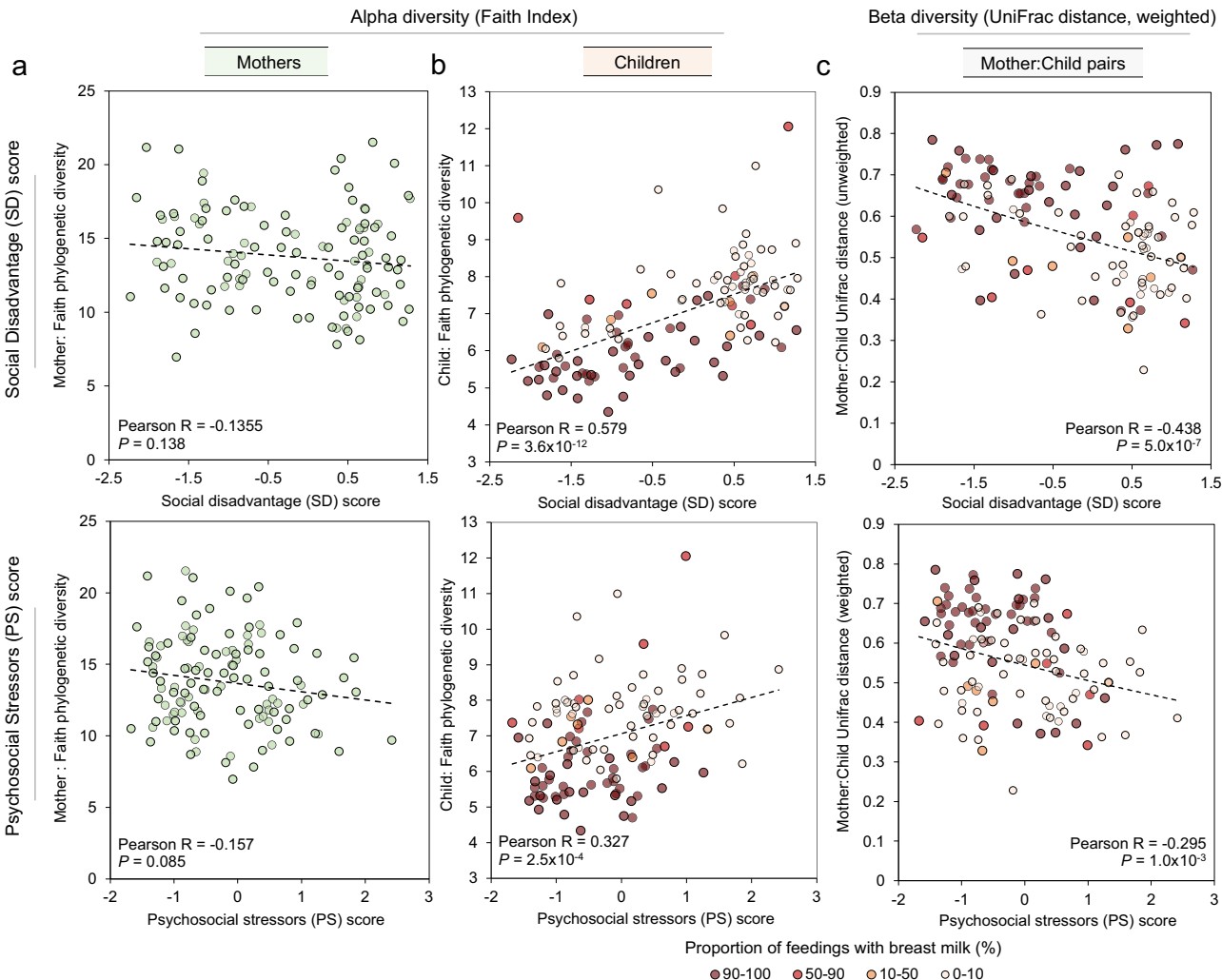

**Fig. 2 | GM sample diversity and composition comparisons with Social Disadvantage (SD) and Psychosocial Stressors (PS) scores, using two-sided T-distribution correlation tests (no adjustment for multiple comparisons).** **a** SD scores and PD scores do not significantly correlate with GM α-diversity (Faith phylogenetic diversity) in the 121 GM samples from the mothers (two-sided T-statistic correlation test). **b** α-diversity in the children is significantly positively correlated with SD and PS scores. Relative proportions of human milk feeding are shown, to provide additional context for potential sources of differential diversity. **c** β-diversity, measured by weighted UniFrac distance between the GM of each mother-child dyad, is positively correlated with SD and PS scores.

"verification" R package, version 1.42; Fig. 5), and as low-PS or high-PS with 72.1% accuracy ($P = 8.2 \times 10^{-4}$, FDR-corrected binomial distribution test; AUC = 0.735, $P = 1.6 \times 10^{-4}$, two-sided Wilcoxon rank sum test calculated using the "roc.area" function in the "verification" R package, version 1.42). The top 25 strongest predictors of "high" vs. "low" SD (Fig. 6a) and PS (Fig. 6b) scores in the mothers were identified according to "mean decrease of accuracy" (MDA) scores for each genome (the average decrease in accuracy by randomly permutating the feature values in "out-of-bag" samples), with additional confidence of differential abundance provided by LEfSe and ANCOM-BC2 analysis shown for each taxa.

The species most associated with low-SD and low-PS in the mothers tended to be detected with zero or very low abundance in the high-SD or high-PS samples (Fig. 6). Low-SD mothers were characterized by increased abundance of many Firmicutes A genomes (Fig. 6a). *Lawsonibacter asaccharolyticus*, a recently identified butyrate-producing species[52,53], was the top predictor of low-SD scores in the mothers (RF MDA = 4.83%, LEfSe LDA effect size=2.7, LEfSe Kruskal-Wallis FDR-corrected $P = 1.8 \times 10^{-5}$, ANCOM-BC2 FDR-corrected $P = 2.9 \times 10^{-4}$). GM-derived butyrate has wide ranging beneficial effects on health, including regulating fluid transport, reducing inflammation, and modulating intestinal motility via mechanisms that include potent

regulation of gene expression[54]. However, *L. asaccharolyticus* has not previously been independently associated with these beneficial effects.

The predictors of high-SD and high-PS in the mothers were phylogenetically distinct, and enriched for Actinobacteria and Firmicutes C, vs. mainly Firmicutes A, respectively. Genomes from four *Bifidobacterium* species (*B. catenulatum*, *B. bifidum*, *B. breve*, and *B. infantis*) are among the seven top predictors of high-SD, and *B. sp002742445* and *B. catenulatum* were also predictors of high-PS. It is recognized that Bifidobacteria in the human gut vary with age, and while quantitatively some are particularly important in the infant GM, its presence in older individuals is generally stable. In general, high abundance of Bifidobacteria is related to gut homeostasis, health maintenance, and protection, in part by producing potentially health-promoting metabolites including short chain fatty acids, conjugated linoleic acid, and bacteriocins, thus Bifidobacteria is postulated to improve health[55]. However, qualitative and quantitative increase of Bifidobacteria are associated with inflammatory disorders (such as diverticulitis, inflammatory bowel disease, and colorectal cancer[56]). Additionally, *Bifidobacterium* is one of three genera most consistently associated with major depressive disorder (MDD) across studies[57]. While the specific functional role of these high-SD associated Bifidobacteria species is

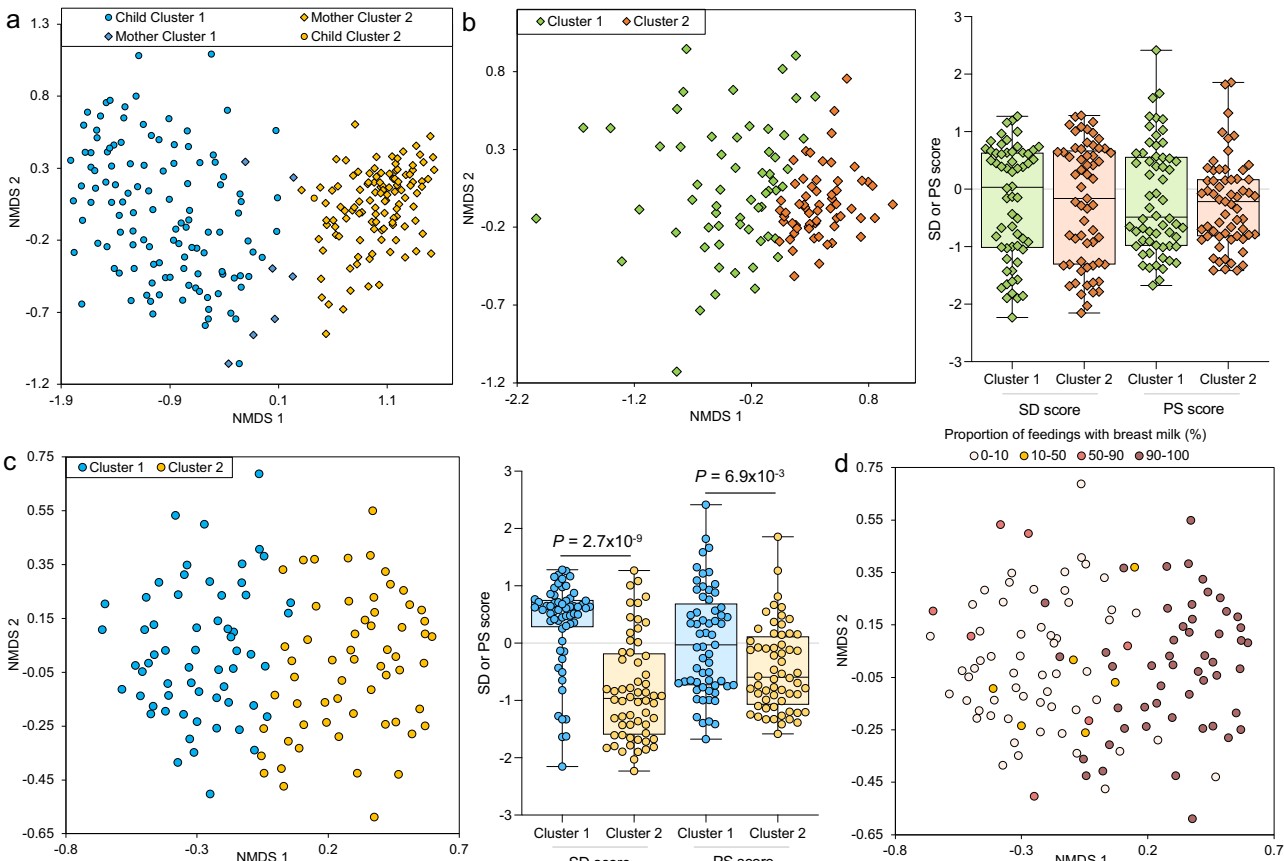

**Fig. 3 | NMDS clustering based on 16 S relative abundance of taxa in the GM, showing major cluster separations according to Dirichlet Multinomial Mixtures (DMM) clustering. a** Mothers and children group into two clusters, almost entirely separating children (cluster 1, 94.5% of cluster members) from mothers (cluster 2, 99.1% of cluster members). **b** Mother samples ($N = 121$ biologically independent samples) grouped into two clusters (n1 = 61 and n2 = 61) but showed no significant differences in social disadvantage (SD) or psychosocial stressors (PS; two-sided Mann−Whitney $U$-tests with FDR adjustment; $P = 0.944$ for each comparison). The range of the boxes extends from the 25th to the 75th percentiles, the whiskers extend from the minimum to the maximum values, and the black horizontal lines in the boxes indicate the mean values of the data. **c** Children samples ($N = 121$ biologically independent samples) also grouped into two clusters (n1 = 61 and n2 = 61). Cluster 1 has significantly higher SD and PS scores (two-sided Mann−Whitney $U$-tests with FDR adjustment). The range of the boxes extends from the 25th to the 75th percentiles, the whiskers extend from the minimum to the maximum values, and the black horizontal lines in the boxes indicate the mean values of the data. **d** The proportion of feedings with breast milk is shown on the NMDS clustering.

unclear, there is a striking increase in overall Bifidobacterial abundance in high-SD mothers (Supplementary Fig. 6).

*Bacteroides A mediterraneensis* was the top predictor of low-PS in mothers (RF MDA = 3.5%, LEfSe LDA effect size=2.5, LEfSe Kruskal-Wallis FDR-corrected $P = 3.8 \times 10^{-4}$; Fig. 6b). In mice, stress exposure reduces *Bacteroides* abundance in the GM[58], and in humans, *Bacteroides* is one of five genera associated with healthy status vs. MDD patients[59]. However, this is the first report of *B. A mediterraneensis* specifically being associated with PS in a human cohort. *Faecalibacterium* and *Prevotella* were also negatively associated with MDD[59], and in our study *Faecalibacterium* sp. and *Prevotella* sp00127S135 were the 6th and 9th strongest predictors of low-PS in the mothers(respectively). *S. thermophilus* was also among the predictors of low-SD, with zero or low abundance in high-PS and high-SD individuals.

Three species of *Blautia* were among the top predictors of high-PS in the mothers, and none were associated with SD, suggesting a specific link with psychosocial stressors. In humans, *Blautia* is one of ten genera associated with MDD[59], and *Blautia* and *Eggerthella* (represented in the high-PS mothers by *E. lenta*) were significantly correlated with PSS scores[60]. The latter study also identified *Blautia* and *Bifidobacteria* (represented in the high-PS mothers by *B. catenulatum*) as being significantly associated with MDD[60]. However, the overall abundance of *Blautia* in the GM was fairly consistent across mothers (Supplementary Fig. 7), and only the three specific species identified in

Fig. 6b predicted PS, highlighting the importance of species-level quantification provided by WMS.

**Bacterial species including members of Enterobacter and Bifidobacterium distinguish the GMs of children based on SD and PS**
We identified discriminatory bacterial genomes that classified children as low-SD and high-SD with 83.3% accuracy ($P = 1.3 \times 10^{-7}$; FDR-corrected binomial distribution test; AUC = 0.868, $P = 2.9 \times 10^{-9}$, two-sided Wilcoxon rank sum test calculated using the "roc.area" function in the "verification" R package, version 1.42; Fig. 5), and as low-PS and high-PS with 67.6% accuracy ($P = 1.3 \times 10^{-7}$; FDR-corrected binomial distribution test; AUC = 0.721, $P = 2.1 \times 10^{-3}$, two-sided Wilcoxon rank sum test calculated using the "roc.area" function in the "verification" R package, version 1.42). The top 25 predictors of SD score (Fig. 7a) and PS score (Fig. 7b) in the children were identified according to MDA scores.

Among the strongest predictors of high-SD in children were *Enterobacter nimipressuralis* (RF MDA = 5.94%, LEfSe LDA effect size=3.4, LEfSe Kruskal-Wallis FDR-corrected $P = 2.3 \times 10^{-6}$; only detected in one low-SD child) and *Klebsiella pneumoniae* (RF MDA = 3.09%, LEfSe LDA effect size=3.6, LEfSe Kruskal-Wallis FDR-corrected $P = 9.9 \times 10^{-5}$), both proinflammatory lipopolysaccharide-expressing Proteobacteria[61]. The strongest predictor of children with low-SD was *B. infantis*, a species frequently used as a probiotic to diminish

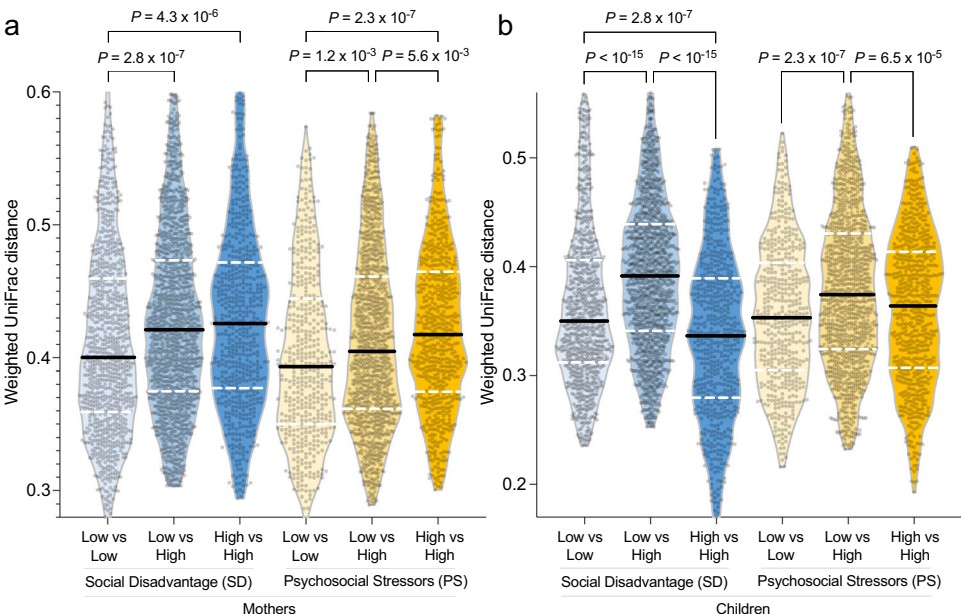

**Fig. 4 | Comparisons of β-diversity between sample sets based on high-vs-low SD and PS, quantified by weighted UniFrac distance between sample pairs and two-sided FDR-corrected Mann–Whitney U-tests to test for significant differences. a** Comparisons of within- and between-group β-diversity of the GM for mothers with low-SD ($n = 35$), high-SD ($n = 43$), low-PS ($n = 36$) and high-PS ($n = 32$), all representing biologically independent samples. **b** Comparisons of within- and between-group GM β-diversity in children with high-SD, low-SD, high-PS and low-PS ($n$-values same as for the mothers). The number of values used for the tests were

the number of unique biologically independent sample pairs in each group ($n = 903$ low-SD vs low-SD, $n = 1505$ high-SD vs low-SD, $n = 595$ high-SD vs high-SD, $n = 496$ low-PS vs low-PS, $n = 1152$ high-PS vs low-PS, $n = 630$ high-PS vs high-PS). On the violin plots, thick black lines indicate the median values, white dashed lines indicate the quartiles of the data range, the width of the shaded areas represent the proportion of data points located at the given weighted UniFrac distance, and the height of the shaded areas spans from the minimum to the maximum value.

inflammation and associated with breastfeeding[62]. *B. infantis* represented an average of 28.3% of the GM in low-SD infants, but only 5.2% in high-SD infants (RF MDA = 5.06%, LEfSe LDA effect size=5.1, LEfSe Kruskal-Wallis FDR-corrected $P = 0.01$; Supplementary Fig. 7). In healthy term infants, low abundance of Bifidobacteriaceae has been linked with systemic inflammation and immune dysregulation early in life[63], development of autoimmunity and early-onset type 1 diabetes[64], as well as neurodevelopment impairment in preterm infants[65]. The microbial differences found in our infants are partly related to variation in breast milk exposure, which is diminished in infants living in high SD/PS (Figs. 2b, c). Beyond breast milk, other lifestyle differences have also been shown to have important impacts on early life GM assembly that persist, intergenerationally and well beyond infancy[66]. This highlights the biologic conversion of social disadvantage, in this case linked to breast milk feeding, with the GM and subsequent potential impact on long term health outcomes.

There was an overlap of the predictors for low-SD and low-PS in the children, including *Veillonella parvula A* and several *Collinsella* spp. *Veillonella* is a signature genus of the 4 month microbiome, and with *Collinsella* was found in the breast-fed GM, indicating reduced oxygen concentration and increased production and utilization of lactic acid, which is specific for a milk dominated diet[67]. The high-SD and high-PS-discriminating bacteria were enriched for a broad range of evolutionarily distinct Firmicutes species (Fig. 7). The best high-PS predictors included *F. gnavus*, a pathobiont associated with inflammatory bowel disease[68] and *Sutterella* sp. *Sutterella* species are prevalent commensals in the human GM and have mild-proinflammatory properties[69].

**Metabolic pathways associated with SD and PS include carbohydrate degradation and L-glutamate and L-glutamine synthesis**
The same statistical approaches (Supplementary Data 1j) were used to identify SD- and PS-discriminatory metabolic pathways (using a curated database of metabolic pathways, MetaCyc[45]). Based on metabolic

pathway profiles (Supplementary Data 2), RF classified mothers as high- and low-SD with 65.4% accuracy ($P = 0.05$; FDR-corrected binomial distribution test; Supplementary Data 1h) and as high- and low-PS with 60.3% accuracy ($P = 0.09$; FDR-corrected binomial distribution test), and classified children as high- and low-SD with 80.8% accuracy ($P = 5.2 \times 10^{-7}$; FDR-corrected binomial distribution test) and as high- and low-PS with 61.8% accuracy ($P = 0.056$; FDR-corrected binomial distribution test). Compared to genomic abundance data, pathway abundance RF accuracy of classification was lower for SD and PS scores because it has fewer total features and greater conservation of detection across samples (Supplementary Fig. 5). However, the MDA values from the RF (using the top 25 pathways) combined with LEfSe P-values still provided a means to identify the metagenomic pathways most associated with high SD and PS (Table 2).

The three pathways with the highest predictive value for high-SD in mothers (Table 2) related to carbohydrate degradation (sucrose degradation IV, glycogen degradation I, starch degradation III), which appears to relate to the accompanying abundance of strict anaerobic *Bifidobacterium* species that are rich in carbohydrate metabolism pathways[70], with *B. bifidum* and *B. infantis* contributing 62.8% of the total abundance of sucrose degradation IV and 56.5% of the total abundance to glycogen degradation I (Fig. 8a). These taxa also contribute to the enriched pathways UDP-N-acetyl-D-glucosamine biosynthesis I and Superpathway of L-threonine biosynthesis, further highlighting their importance not only to the overall SD-defining taxonomic profile, but also to the metabolic potential of the GM. The "myo-, chiro- and scyllo-inositol degradation" pathway (PWY-7237) was among the most strongly associated with high-PS in mothers (RF MDA = 2.57%, LEfSe LDA effect size=1.5, LEfSe Kruskal-Wallis FDR-corrected $P = 7.4 \times 10^{-3}$). Myo-inositol and chiro-inositol degradation by the GM contributes to inositol deficiency[71], which includes metabolic disorders involved with insulin function[71] and MDD when myo-inositol is deficient in the prefrontal cortex[72].

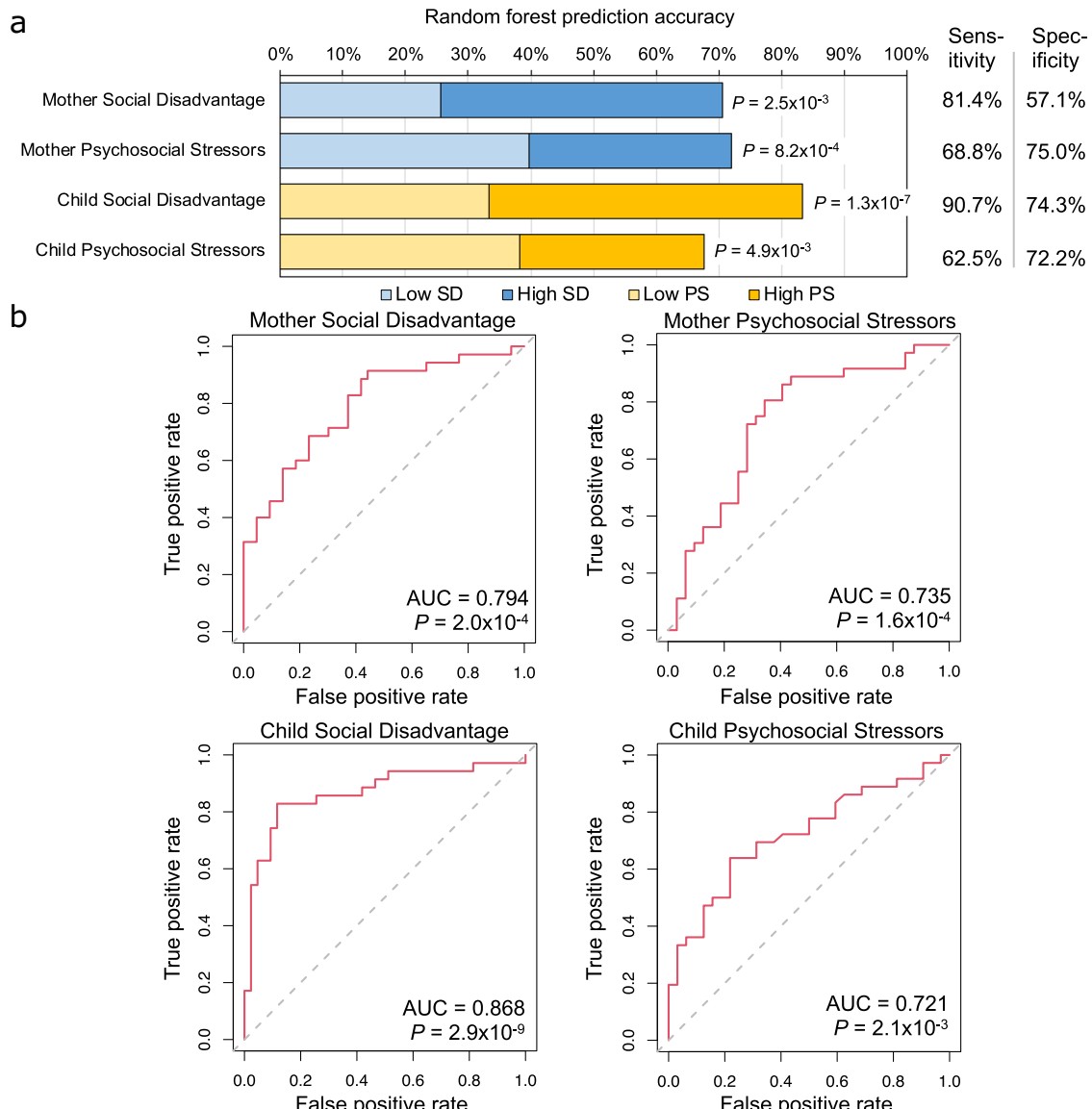

**Fig. 5 | The Random Forest (RF) classification accuracy (low-SD vs high-SD and low-PS vs high-PS, out-of-bag error) based on relative UHGG genome abundance in the mothers and children. a** Overall classification accuracy, with *P*-values indicating significance based on FDR-corrected binomial distribution tests (compared to random sample assignment). **b** For each of the four classification tests, receiver operating characteristic (ROC) curves are shown based on RF models, with the area under the curve (AUC) scores and associated two-sided Wilcoxon rank sum test results for each ROC curve indicated.

The top predictive pathways of SD in children (Table 2) included L-glutamate and L-glutamine synthesis (PWY-5505) as being second-most strongly associated with high-SD in the children (RF MDA = 5.01%, LEfSe LDA effect size=1.2, LEfSe Kruskal-Wallis FDR-corrected $P = 9.5 \times 10^{-6}$), which was previously associated with obesity and visceral fat accumulation[73]. Here, L-glutamine biosynthesis was also associated with high-SD in both mothers and children. In supplementation studies of the GM, glutamine reduces the ratio of Firmicutes to Bacteroidetes and bacterial overgrowth or bacterial translocation and increases the density of secretory immunoglobulin A (IgA) and IgA+ cells in the intestinal lumen[74]. The top metabolic pathway associated with high SD in children relates to tryptophan biosynthesis. Tryptophan is the sole precursor of the neurotransmitter serotonin as well as other active metabolites. Dysregulation of tryptophan metabolism is emerging as having a potential role in neurologic function and psychiatric disorders[75–77]. The facultative anaerobe *Klebsiella pneumonia* (the fourth-ranked taxa associated with high SD in the children; Fig. 7a) was responsible for (i) 30% of the total abundance of the top pathway

(pyrimidine deoxyribonucleotides biosynthesis from CTP) and (ii) 33% of the total abundance of the fourth-ranked pathway, superpathway of glycerol degradation to 1,3-propanediol, a function first described in and well-studied in *Klebsiella pneumonia*[78], although the function of this pathway in the gut microbiome is unclear (Fig. 8b).

**Bacterial species discrimination based on maternal circulating cytokines**

The relationship between inflammation and the GM was examined by measuring maternal circulating cytokines IL-6, IL-8, IL-10, and TNFα at the third trimester. For all circulating cytokines, the average concentrations were not significantly different between the low-SD and high-SD participants or between the low-PS and high-PS groups (Supplementary Data 1a). The only circulating cytokine with any significant overall correlation with SD or PS was IL-8 ($r = 0.252$, degrees of freedom=81, $P = 0.022$, effect size statistic=2.34, 95% confidence interval = −1.98:1.98, two-sided-test; Supplementary Fig. 8, Supplementary Data 1f). The same RF approach used to compare high-vs-low

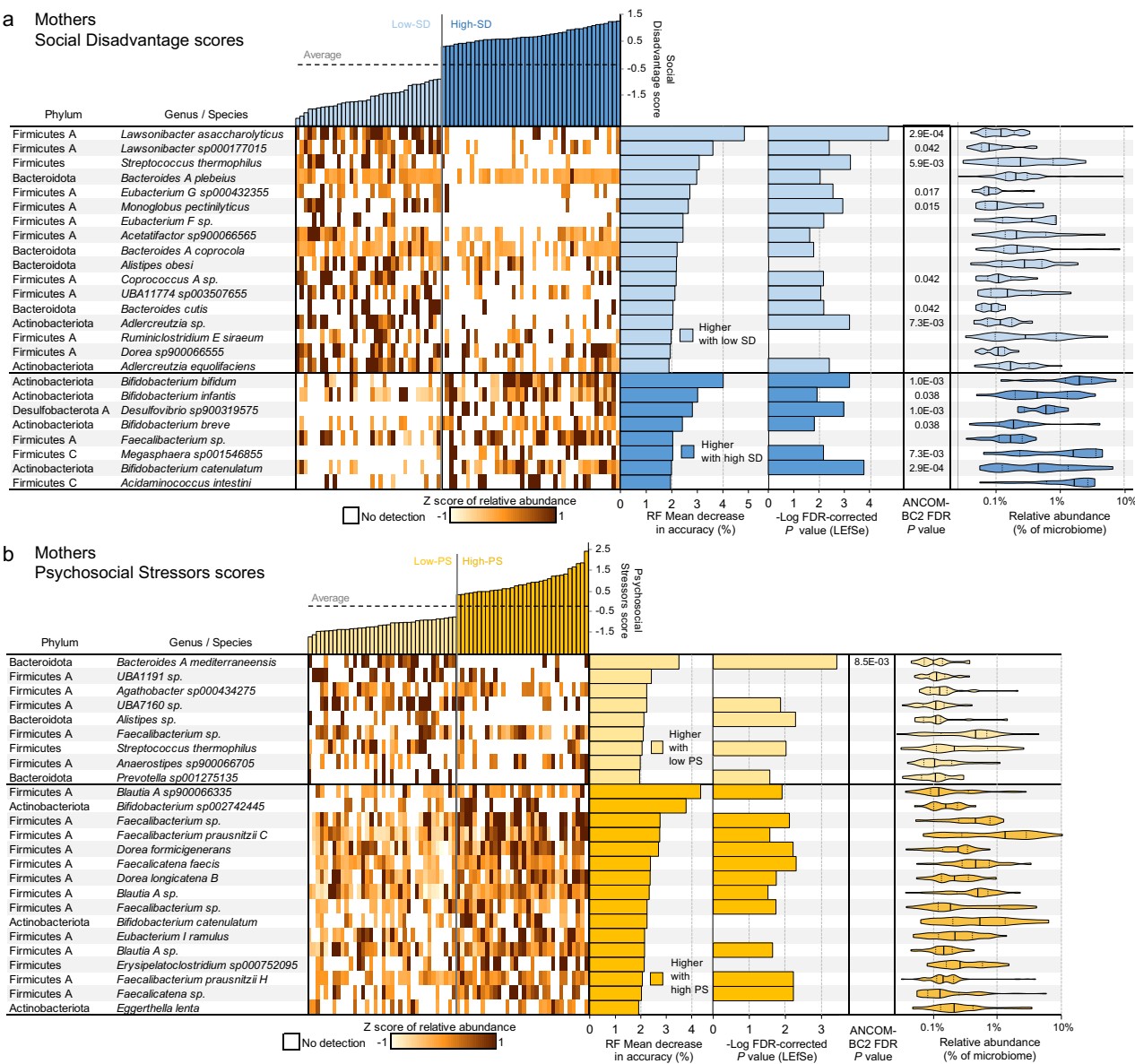

**Fig. 6 | WMS genome differential abundance in the mothers, based on comparisons of high-vs-low Social Disadvantage (SD) scores ($N = 35$ and $N = 41$ biologically independent samples, respectively) and high-vs-low Psychosocial Stressors (PS) scores ($N = 36$ and $N = 32$ biologically independent samples, respectively). a** Taxonomy and relative abundance per sample for the genomes with the highest predictive value for SD in the random forest (RF) model based on the GMs from the mothers (ranked by mean decrease in accuracy of the RF model; MDA). Also displayed are -Log of the Kruskal-Wallis test FDR-corrected *P*-values from LEfSe (when effect size ≥ 2), FDR-corrected *P*-values from ANCOM-BC2 and

the relative abundance of the taxa when present. On violin plots, thick black lines indicate median values, dashed black lines indicate data range quartiles, shaded area heights represent the proportion of data points for the abundance values, and the width of the shaded areas spans from the minimum to the maximum value. Violin plots represent data from all samples in the comparison. **b** Taxonomy, relative abundance and differential abundance statistics for the genomes with the highest predictive value for PS in the RF model based on the GMs from the mothers. Data visualization and statistics are the same as for **a**.

SD and PS was used to identify the classification accuracy of high-vs-low circulating cytokines, in order to test if the metagenomic profiles of children or mothers are distinct between cohorts based on cytokine levels. After dividing groups using the same approach ("high" and "low" defined as being greater than or less than the average value + 0.5 and −0.5 standard deviations, respectively), the sample sizes were substantially smaller for the cytokine comparisons (IL-6, $N = 45$; IL-8, $N = 50$; IL-10, $N = 51$, TNF-α, $N = 55$), compared to SD ($N = 77$) and PS ($N = 68$). This is due to (i) The participants not being selected based on the extremes of cytokine values, as they were for SD and PS, resulting in most samples have close to average values (see Supplementary

Fig. 8) and (ii) Cytokine values not being available for six of the participants.

Performing the RF comparison for cytokines using genomes and pathways, only the children's genomic profiles were significantly predictive of only the high-vs-low IL-6 status in the mother (66.7% accuracy, $P = 0.029$ by FDR-corrected binomial distribution test); Supplementary Table 1; Statistical results for this comparison are in Supplementary Table SI. In preclinical models, IL-6 is centrally important in altering fetal brain development in maternal immune activation models, where placental inflammatory signals are relayed to the fetal brain[79–81]. In this cohort, the top three genomes associated

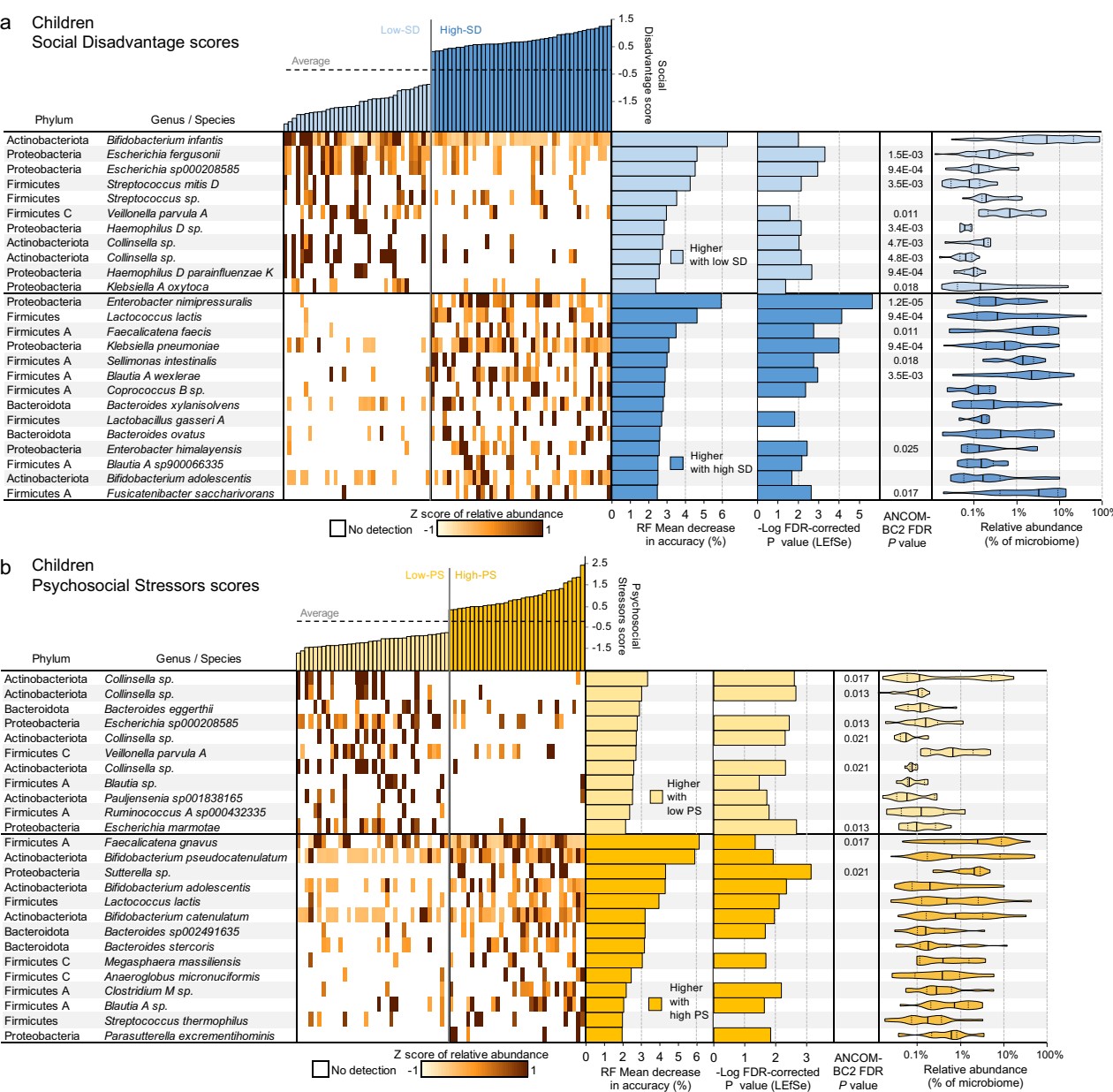

**Fig. 7 | WMS genome differential abundance in the children, based on comparisons of high-vs-low Social Disadvantage (SD) scores (_N_ = 35 and _N_ = 41 biologically independent samples, respectively) and high-vs-low Psychosocial Stressors (PS) scores (_N_ = 36 and _N_ = 32 biologically independent samples, respectively). a** Taxonomy and relative abundance per sample for genomes with the highest predictive value for SD in the random forest (RF) model based on the GMs from the children (ranked by mean decrease in accuracy of the RF model; MDA). Also shown are -Log of the Kruskal-Wallis test FDR-corrected _P_-values from LEfSe (when effect size ≥ 2), FDR-corrected _P_-values from ANCOM-BC2 and the relative abundance of the taxa when present. On violin plots, thick black lines indicate median values, dashed black lines indicate data range quartiles, shaded area heights represent the proportion of data points for the abundance values, and the width of shaded areas spans from the minimum to the maximum value. Violin plots represent data from all samples in the comparison. **b** Taxonomy, relative abundance and differential abundance statistics for the genomes with the highest predictive value for PS in the RF model based on the GMs from the children. Data visualization and statistics are the same as for **a**.

with low IL-6 are (i) _B. bifidum_ (RF MDA = 4.48, LEfSe LDA effect size=4.44, LEfSe Kruskal-Wallis FDR-corrected _P_ = 0.044, ANCOM-BC2 FDR-adjusted _P_ = 0.037), which was the top genome associated with high-SD in the mothers (Fig. 6), (ii) C _ollinsella_ sp. (RF MDA = 3.46, LEfSe LDA effect size=2.52, LEfSe Kruskal-Wallis FDR-corrected P = 0.044, ANCOM-BC2 FDR-adjusted _P_ = 0.045), which was strongly associated with low-PS in the children (the top two genomes and four of the top seven genomes in that comparison; Fig. 7), and (iii) _B. infantis_ (RF MDA = 3.29, LEfSe LDA effect size=4.70, LEfSe Kruskal-Wallis FDR-corrected _P_ = 0.044, ANCOM-BC2 FDR-adjusted _P_ = 0.032), associated with low-SD in the children and high-SD in the mothers.

Although the overall RF model for the pathways did not show significant predictive accuracy in any comparison (Supplementary Table 1), the same pathway identification approach shown in Table 2 was used to identify the top pathways associated with high and low IL-6 in the children (results in Supplementary Data 1j). The top child pathways associated with low maternal IL-6 include glycogen degradation I (GLYCOCAT-PWY; MDA = 1.71, LEfSe LDA effect size=1.90, LEfSe Kruskal-Wallis FDR-corrected _P_ = 0.018), which was among the top associated with high-SD mothers (Fig. 6) and came primarily from _B. bifidum_ and _B. infantis_ (Fig. 8), as well as the related pathway glucose and glucose-1-phosphate degradation (GLUCOSE1PMETAB-PWY, RF

**Table 2 | Metabolic pathways differential abundance in each of the RF comparisons**

| Comparison | MetaCyc pathway ID | Average abundance (CPM) | | | Random Forest MDA (%) | LEfSe results | |
|---|---|---|---|---|---|---|---|
| | | Low SD/PS | High SD/PS | Direction | | LDA Effect size | K-W test FDR value |
| Mother - Social Disadvantage (SD) | PWY-7198: pyrimidine deoxyribonucleotides de novo biosynthesis IV | 61.9 | 100.9 | ↑ | 5.38 | 1.30 | $9.6×10^{-5}$ |
| | PWY-5384: sucrose degradation IV (sucrose phosphorylase) | 31.2 | 61.8 | ↑ | 3.78 | 1.19 | $5.7×10^{-4}$ |
| | PWY-6549: L-glutamine biosynthesis III | 35.2 | 60.1 | ↑ | 3.65 | 1.13 | $9.6×10^{-5}$ |
| | GLYCOCAT-PWY: glycogen degradation I | 38.7 | 72.7 | ↑ | 3.01 | 1.21 | $4.7×10^{-3}$ |
| | UDPNAGSYN-PWY: UDP-N-acetyl-D-glucosamine biosynthesis I | 172.4 | 208.8 | ↑ | 2.91 | 1.27 | 0.018 |
| | THRESYN-PWY: superpathway of L-threonine biosynthesis | 313.6 | 339.2 | ↑ | 2.67 | 1.17 | 0.046 |
| | PWY-241: C4 photosynthetic carbon assimilation cycle, NADP-ME type | 45.8 | 84.0 | ↑ | 2.20 | 1.30 | $5.9×10^{-4}$ |
| | PWY-6901: superpathway of glucose and xylose degradation | 106.1 | 138.2 | ↑ | 2.04 | 1.21 | $1.5×10^{-3}$ |
| | PWY-5913: partial TCA cycle (obligate autotrophs) | 42.4 | 85.9 | ↑ | 1.86 | 1.35 | $4.7×10^{-4}$ |
| | PWY-622: starch biosynthesis | 26.9 | 64.7 | ↑ | 1.82 | 1.28 | $5.7×10^{-4}$ |
| | PWY-6731: starch degradation III | 22.3 | 42.3 | ↑ | 1.73 | 1.00 | $1.9×10^{-3}$ |
| | NAGLIPASYN-PWY: lipid IVA biosynthesis (E. coli) | 80.9 | 58.2 | ↓ | 3.04 | 1.10 | 0.022 |
| Mother - Psychological Stressors (PS) | ANAGLYCOLYSIS-PWY: glycolysis III (from glucose) | 392.8 | 427.7 | ↑ | 4.52 | 1.27 | $3.2×10^{-3}$ |
| | GLYCOGENSYNTH-PWY: glycogen biosynthesis I (ADP-D-Glucose) | 297.0 | 382.7 | ↑ | 4.16 | 1.60 | $2.6×10^{-3}$ |
| | RHAMCAT-PWY: L-rhamnose degradation I | 140.9 | 168.7 | ↑ | 3.48 | 1.22 | $8.9×10^{-3}$ |
| | VALSYN-PWY: L-valine biosynthesis | 493.6 | 540.3 | ↑ | 2.74 | 1.36 | $8.9×10^{-3}$ |
| | THISYNARA-PWY: superpathway of thiamine diphosphate biosyn. III | 214.5 | 242.2 | ↑ | 2.69 | 1.14 | 0.017 |
| | PWY-7237: myo-, chiro- and scyllo-inositol degradation | 225.9 | 285.6 | ↑ | 2.57 | 1.49 | $7.4×10^{-3}$ |
| | PWY-6731: starch degradation III | 24.0 | 42.9 | ↑ | 2.56 | 1.01 | $8.9×10^{-3}$ |
| | PWY-7357: thiamine phosphate formation from pyrithiamine &oxythiamine | 283.5 | 333.4 | ↑ | 2.41 | 1.38 | $5.5×10^{-3}$ |
| | NONOXIPENT-PWY: pentose phosphate pathway (non-oxidative branch) I | 252.0 | 300.2 | ↑ | 2.33 | 1.36 | $9.8×10^{-3}$ |
| | PWY-622: starch biosynthesis | 34.1 | 61.1 | ↑ | 2.12 | 1.14 | $8.9×10^{-3}$ |
| | PWY-7115: C4 photosynthetic carbon assimilation cycle, NAD-ME type | 43.4 | 63.2 | ↑ | 2.02 | 1.01 | 0.022 |
| | DTDPRHAMSYN-PWY: dTDP-β-L-rhamnose biosynthesis | 539.2 | 574.2 | ↑ | 2.01 | 1.26 | 0.026 |
| | COMPLETE-ARO-PWY: superpathway of aromatic amino acid biosynthesis | 435.5 | 470.6 | ↑ | 1.93 | 1.22 | $8.9×10^{-3}$ |
| | GLUTORN-PWY: L-ornithine biosynthesis I | 307.3 | 359.2 | ↑ | 1.91 | 1.38 | $8.9×10^{-3}$ |
| Child - Psychological Stressors (PS) | PWY-7210: pyrimidine deoxyribonucleotides biosynthesis from CTP | 29.2 | 78.8 | ↑ | 5.08 | 1.42 | $6.8×10^{-4}$ |
| | PWY–5505: L-glutamate and L-glutamine biosynthesis | 9.3 | 38.7 | ↑ | 5.01 | 1.20 | $9.5×10^{-6}$ |
| | PWY-6897: thiamine diphosphate salvage II | 167.3 | 223.2 | ↑ | 4.01 | 1.43 | $3.8×10^{-4}$ |
| | GOLPDLCAT-PWY: superpath. of glycerol degradation to 1,3-propanediol | 35.1 | 58.2 | ↑ | 3.48 | 1.00 | $1.7×10^{-3}$ |
| | PWY0-1296: purine ribonucleosides degradation | 205.7 | 255.8 | ↑ | 3.46 | 1.39 | 0.032 |
| | PWY-6549: L-glutamine biosynthesis III | 33.9 | 58.5 | ↑ | 3.16 | 1.12 | $3.8×10^{-4}$ |
| | PWY-6470: peptidoglycan biosynthesis V (β-lactam resistance) | 26.6 | 74.8 | ↑ | 3.03 | 1.34 | $3.8×10^{-4}$ |
| | PWY-6124: inosine-5′-phosphate biosynthesis II | 275.7 | 341.1 | ↑ | 2.73 | 1.54 | $6.8×10^{-4}$ |
| | PWY-6700: queuosine biosynthesis I (de novo) | 207.5 | 292.3 | ↑ | 2.72 | 1.61 | $8.1×10^{-4}$ |
| | TEICHOICACID-PWY: poly(glycerol phosphate) wall teichoic acid biosyn. | 21.8 | 48.9 | ↑ | 2.59 | 1.15 | $5.1×10^{-4}$ |
| | PWY-1042: glycolysis IV | 396.4 | 483.2 | ↑ | 2.54 | 1.60 | $1.7×10^{-3}$ |
| | PWY0–1479: tRNA processing | 187.7 | 132.5 | ↓ | 3.74 | 1.46 | $1.8×10^{-3}$ |
| Child - Social Disadvantage (SD) | TRPSYN-PWY: L-tryptophan biosynthesis | 266.0 | 321.2 | ↑ | 3.39 | 1.43 | 0.015 |
| | PWY-6549: L-glutamine biosynthesis III | 29.2 | 56.9 | ↑ | 3.25 | 1.15 | $6.5×10^{-4}$ |
| | PWY-241: C4 photosynthetic carbon assimilation cycle, NADP-ME type | 83.4 | 119.1 | ↑ | 2.48 | 1.26 | 0.015 |
| | ANAEROFRUCAT-PWY: homolactic fermentation | 281.5 | 331.0 | ↑ | 2.43 | 1.39 | 0.015 |
| | PWY–5505: L-glutamate and L-glutamine biosynthesis | 13.5 | 37.6 | ↑ | 2.37 | 1.13 | $7.4×10^{-4}$ |

**Table 2 (continued) | Metabolic pathways differential abundance in each of the RF comparisons**

| Comparison | MetaCyc pathway ID | Average abundance (CPM) | | | Random Forest MDA (%) | LEfSe results | |
|---|---|---|---|---|---|---|---|
| | | Low SD/PS | High SD/PS | Direction | | LDA Effect size | K-W test FDR value |
| | PANTOSYN-PWY: superpathway of coenzyme A biosynthesis I (bacteria) | 230.8 | 281.5 | ↑ | 2.34 | 1.43 | 0.024 |
| | PWY–4981: L-proline biosynthesis II (from arginine) | 68.9 | 115.6 | ↑ | 2.10 | 1.42 | 0.015 |
| | PWY-6731: starch degradation III | 59.1 | 86.1 | ↑ | 1.99 | 1.16 | 0.034 |
| | FUC-RHAMCAT-PWY: superpathway of fucose and rhamnose degradation | 76.4 | 53.8 | ↓ | 2.36 | 1.08 | 0.024 |

"↑" indicates pathways higher with high Social Disadvantage (SD) / Psychosocial Stressors (PS), and "↓" indicates pathways higher with low SD/PS.
Average abundance and association (high or low) are shown for the top 25 pathways with the highest predictive value in the RF model (ranked by mean decrease in accuracy of the RF model; MDA), and with LEfSe LDA effect size ≥ 1 and FDR-corrected Kruskal–Wallis test ≤ 0.05.

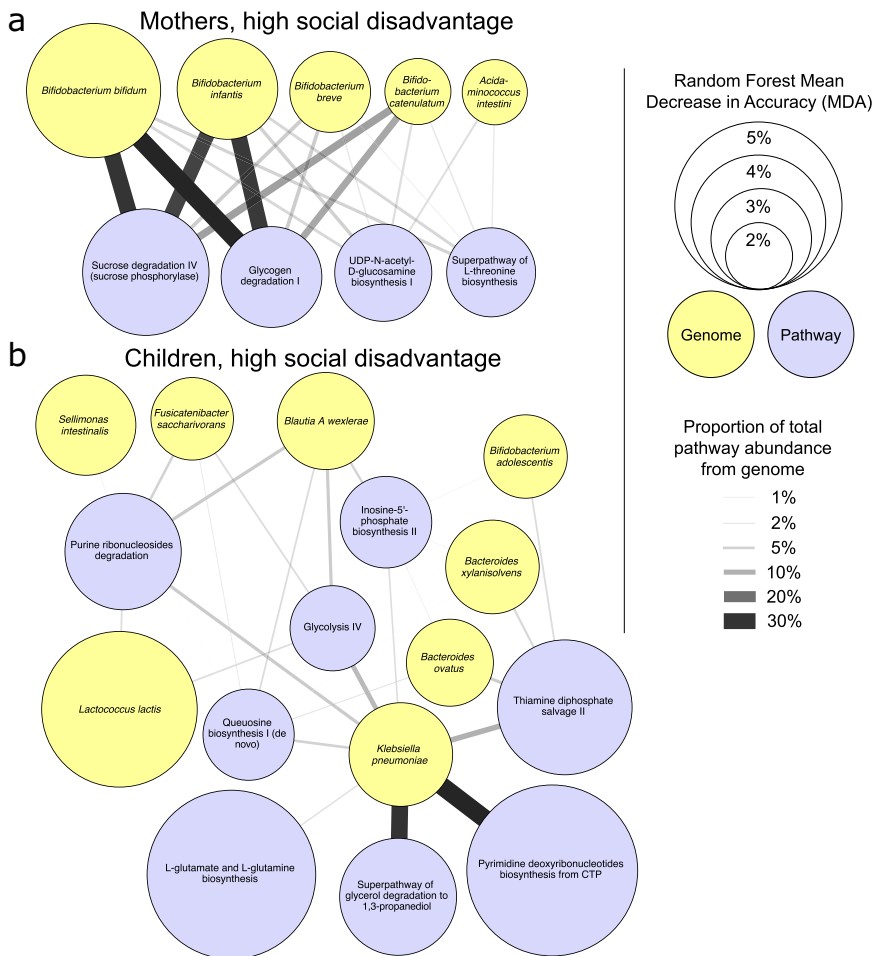

**Fig. 8 | Networks depicting the relative contribution of high-social disadvantage (SD)-associated genomes to high-SD associated pathways. a** Edges connect high-SD genomes in the mothers (yellow, from Fig. 6) to high-SD pathways in the mothers (purple, from Table 2). **b** Edges connect high-SD genomes in the children (yellow, from Fig. 7) to high-SD pathways in the children (purple, from Table 2). Node sizes indicate the mean decrease in accuracy (MDA) values from the random forest analysis, and edge thickness and darkness indicate the proportion of reads that each genome contributes to the pathways.

MDA = 2.53, LEfSe LDA effect size=1.87, LEfSe Kruskal-Wallis FDR-corrected P = 0.03). Taken together, these genome and pathway results indicate that the child genomes and pathways distinguishing low maternal IL-6 overlap considerably with the genomes and pathways identifying high-SD in the mothers, although there is no significant correlation between IL-6 and SD (Supplementary Fig. 8).

*B. adolescentis* was the top associated taxa with high maternal IL-6 in the children (RF MDA = 3.99, LEfSe LDA effect size=3.36, LEfSe Kruskal-Wallis FDR-corrected P = 0.038, not significant by ANCOM-BC2), followed by *B. pseudocatenulatum* (RF MDA = 3.81; Not

significant by LEfSe or ANCOM-BC2). Both of these genomes were among the top four predictive of high-PS in the children. The third-ranked genome was *Enterobacter himalayensis* (RF MDA = 3.59, LEfSe LDA effect size=3.04, LEfSe Kruskal-Wallis FDR-corrected P = 0.032, not significant by ANCOM-BC2), which was associated with high-SD in the mothers (Fig. 6). Additionally, six *Blautia* genomes were among the top 13 associated with high-IL-6, a genus which was strongly associated with high-PS in the mothers (Fig. 6). Among the 7 pathways significantly associated with high-IL-6 children were (i) myo-, chiro- and scyllo-inositol degradation (PWY-7237; MDA = 3.6, LEfSe LDA effect

size=1.81, LEfSe Kruskal-Wallis FDR-corrected $P = 3.6 \times 10^{-4}$), which was associated with high-PS mothers, (ii) L-glutamine biosynthesis III (PWY-6549: L-glutamine biosynthesis III; MDA = 2.28, LEfSe LDA effect size=1.26, LEfSe Kruskal-Wallis FDR-corrected $P = 3.3 \times 10^{-3}$), associated with high-SD mothers and children and high-PS children and (iii) thiamine diphosphate salvage II (PWY-6897; MDA = 2.47, LEfSe LDA = 1.38, LEfSe Kruskal-Wallis FDR-corrected $P = 2.7 \times 10^{-3}$), associated with high-PS children. Together, these results indicate that many of the same taxa and pathways that distinguish high maternal IL-6 from the child GM were also predictive of high-PS in the mothers and children.

The human GM modulates inflammatory cytokine production[8,82] and has been linked to chronic inflammatory disorders[6]. Elevated IL-6 concentrations have been linked to specific GM profiles in disease states among adults[83,84] and recently to cerebellar neuropathology with loss of Purkinje cells when elevated during pregnancy in animal models[80,81]. Here, we have identified that many of the genomes and pathways significantly associated with low and high maternal IL-6 intersect with those associated with high-SD in the mothers and high-PS in the mothers and children (respectively).

In conclusion, from our prospectively assembled eLABE cohort, we identified a subset of 121 mother infant dyads (Supplementary Fig. 1) that enabled us to quantify, for the first time, the association of exposure with both SD and PS on GM structure and function for mothers and their infants. This is the first human study to separate the association of SD from PS on the GM, demonstrating distinct taxonomy and functions consistent with divergent underlying drivers for each. This is in agreement with preclinical data demonstrating different causal organisms involved in stress responses related to the hypothalamic pituitary axis[85] compared to environmental exposures like diet[86]. The GMs of mothers and infants classified as "high" (case) or "low" (control) SD and PS show distinct discriminatory taxonomic and metabolic features that accurately 'predict' maternal prenatal SD and PS status, with SD having greater predictive accuracy (81.4% and 90.7% in mothers and children respectively) than PS (68.8% and 62.5%, respectively). Mothers with high-SD/high-PS had highly variable microbiomes compared to low-SD/low-PS mothers, reflecting greater permutations from environmental influences. The distinct taxonomic and functional predictors for SD compared to PS indicate unique underlying mechanisms driving these relationships. Furthermore, we identified a modest but significant relationship between the infant GM and prenatal circulating IL-6 concentrations in mothers (66.7% accuracy). The discriminating taxa for IL-6 were distinct from those for SD/PS, indicating there may be different pathways contributing to chronic inflammation than those identified in the SD/PS results (Table 2). We remain cautious in interpreting these results however, related to limited maternal serum values ($n = 45–55$ samples/group) for these cytokines (methods).

Our work highlights the previously reported negative relationships between socioeconomic status and breastfeeding rates with a subsequent impact on the neonatal GM[87,88]. The resulting GM of highly SD infants at 4 months have greater diversity and a relative paucity of strict anaerobes (*Bifidobacterium, Veillonella* and *Collinsella* sp.) reflective of formula feeding. Evidence is accumulating that these early microbial patterns contribute to select immune mediated long term health outcomes over-represented in SD populations including asthma[89], and diabetes[64]. This highlights the biologic conversion of social disadvantage, in this case linked to breast milk feeding, with the GM and subsequent potential impact on long term health outcomes. These results highlight the need for health policy interventions aimed at increasing breastfeeding resources for mothers with high SD[12].

Our findings should be interpreted in the context of the strengths but also limitations of the study. The findings are associations, and we cannot infer causality. Nonetheless, we identified species-level taxa and metabolic pathways through WMS in human cohort, which forms the basis to test mechanistic causation in preclinical models, which can then be brought back to humans. The population we studied is from a circumscribed demographic region but contains a broad range of socioeconomic backgrounds. Results from cytokine analysis was limited by smaller samples size ($N = 45–55$) and require validation in additional cohorts. We did not examine the role of race because of the collinearity of race and SD, with no additional contribution of race in the model beyond that found with SD alone.

The results identify unique features of the maternal and infant GMs in relation to SD and PS at the time of sampling. Longitudinal studies are needed to determine the stability of these GM signatures from early life to early childhood and will enable elucidating possible enduring effects of socioeconomic status and mental health determinants on GM health and stability. More critically, information on causal pathways triggered or sustained by the GM that affect child health and development could lead to new diagnostic biomarkers and therapeutic interventions. The potential malleability of the GM leaves room for optimism that unfavorable neurodevelopment outcomes might not be inevitable in children to mothers who are experiencing high levels of SD and PS.

## Methods

### Study design, clinical cohort, and ethics statement

This prospective case-control study draws from the Early Life Adversity and Biological Embedding (eLABE[30]) study, with pregnant women identified from the March of Dimes (MOD) Prematurity Research Center at Washington University in St. Louis. Pregnant women ($N = 395$) were recruited between 2017–2020 and delivered at Barnes Jewish Hospital in St. Louis[31] and were recruited by study personnel during clinic visits. One clinic primarily serves patients with public health insurance, and the other primarily serves patients with private health insurance. Eligibility for MOD enrollment was not restrictive, and included plans to deliver at Barnes Jewish Hospital, a minimum age of 18 years, and English speaking. Exclusion criteria included multiple gestations, congenital malformations and infections, premature birth (<37 weeks gestation), maternal alcohol or drug use during pregnancy (excluding tobacco, marijuana), and maternal steroid exposure (excluding inhaled). Race and ethnicity were based on maternal self reporting extracted from the medical record. A priori eligibility for this study at 4 months included if mothers had delivered at ≥37 weeks and infants had reached 4 months corrected age ($N = 355$), with both maternal 3rd trimester (T3) and infant 4 months stool samples available (Supplementary Fig. 1). Mothers had no record of recent antibiotic usage as of the sampling point at T3, and just four of the children had received antibiotics before the 4 month sampling timepoint (two receiving amoxicillin within the first 4 weeks, one receiving azithromycin at 26 days, and one continuously receiving PCN-V; Supplementary Data 2). Women facing social disadvantage were oversampled by increasing recruitment for clinics serving low income women, in order to facilitate the intended comparisons based on social disadvantage and psychological stressors. It is possible that the criteria for English speaking only applicants, limits generalizability and increases bias toward a higher Socioeconomic status. Social Disadvantage however was evenly spread across the cohort (Fig. 1). The requirement of providing a third trimester stool sample may also bias toward a higher socioeconomic status with resources to retrieve and return samples. We attempted to mitigate this with the use of a courier system that retrieved home produced stool samples for all mothers, at all times, to accommodate variable work hours.

All relevant ethical regulations were followed, and the study was approved by the Washington University in St. Louis Institutional Review Board in the Human Research Protection Office (protocol number 201703145), with informed consent obtained from the mother for themselves and their infants. Participants were compensated $20.00 for the stool samples and $25.00 for the blood draws. The

study was performed in accordance with Strengthening the Reporting of Observational Studies in Epidemiology (STROBE) guidelines[90]. Detailed metadata for all samples are provided in Supplementary Data 2 and are summarized for each comparison in Supplementary Data 1a.

## Maternal measures

At each trimester of pregnancy, measures of maternal depression, experiences of stress, as well as demographic and clinical information including insurance, education, address, household composition, pre-pregnancy BMI, and route of delivery were obtained from participants or extracted from the medical record by trained staff. Sex was based on self-reporting, with all mothers reporting as female and the mothers reporting on the behalf of the children.

Components of the two latent factors maternal SD and maternal PS as previously described[30]. Briefly, the components of each latent factor included:

Maternal SD: Insurance status verified in the 3rd trimester from the medical record and maternal self report; Income to Needs Ratio in each trimester based on self-reported family income and houshold size (1.0 being the poverty line for the U.S.); highest self-reported maternal educational level; Area Deprivation Index, a national multidimensional geotracking method based on census block data providing percentile rankings of neighborhood disadvantage status[91]; and maternal nutrition over the past year categorized using the validated Healthy Eating Index (using National Cancer Institute. The Healthy Eating Index−Population Ratio Method. Updated December 14, 2021[92]) obtained using the Diet History Questionnaire (DHQII).

Maternal Psychosocial Stressors: The Edinburgh Postnatal Depression Scale (EPDS)[93] completed in each trimester; Perceived Stress scale (PSS)[94] completed in each trimester and averaged over trimesters; and a one-time lifetime STRAIN survey[95], a comprehensive measure of lifetime stressful and traumatic life events. Experiences of discrimination based on race were assessed using the Everyday Discrimination Scale[96].

## Infant measures

Gestational age was determined by the best obstetric estimate using last menstrual period or earliest ultrasound dating. Birthweight and route of delivery were extracted from the electronic medical record delivery note. Breastfeeding data were collected by parental report at the time of home stool sample collection and based on the Center for Disease Control Infant Feeding Practices II study food frequency checklist data[97,98]. The sex of each child is provided in Supplementary Data 2.

## Biological specimen collection and processing

Maternal serum samples were collected in each trimester, processed within 12 h of collection, and stored at −80°C (details in[31]). Sponta-neously generated stool (feces) from mothers and infants were col-lected from home using a community-based courier system available 24 h per day and stored at −80°C as previously described[98]. All samples were processed in the laboratory of Dr. Phillip Tarr, where DNA was extracted from stools that had been frozen at −80 °C since acquisition using the Qiagen (Hilden, Germany) QIAamp Power Fecal Pro DNA Kit (catalog #51804), and the automated QIAcube[99] (Qiagen). Briefly, 100 mg of stool was suspended in 1.2 mL of stool lysis buffer in a 2 mL screw cap tube containing a mix of 6–8 zirconium beads (2.3 mm, RPI Corporation, Mount Prospect, IL) and 0·1 mL of acid washed glass beads (0.4−0.6 mm, Sigma, MO). This suspension was homogenized by bead beating (FastPrep 24, MP Biomedical, Santa Ana, CA) (6.0 set point, 2 min), and centrifuged (14,000 × g, 3 min, room temperature). 350 μL of clear supernatant was then loaded onto the QIACube rotor adapter for automated DNA purification. The supernatant was then treated on board with Inhibitor Removal Technology (IRT(Qiagen)), to remove inhibitors of subsequent enzymatic. DNA was eluted in 200 μL volume.

## Targeted GM profiling using V4-16S rRNA sequencing, data processing, and analyses

DNA extracted from stool was sequenced on an Illumina MiSeq, pro-ducing 2x250bp paired-end reads spanning the V4 hypervariable region for 242 samples (121 samples from mothers and 121 from their matched children). 16 S sequencing was performed by the Genome Technology Access Center at McDonnell Genome Institute (GTAC@MGI) at Washington University in St. Louis School of Medi-cine, USA. Primer sequences used were forward primer (5′- AATGA-TACGGCGACCACCGAGATCTACACATCGTACGTCGTCGGCAGCGTCA GATGTGTATAAGAGACAGANNNNNGTGCCAGCMGCCGCGGTAA-3′) and reverse primer (5′-CAAGCAGAAGACGGCATACGAGATACCTACT GGTCTCGTGGGCTCGGAGATGTGTATAAGAGACAGNNANNNGGACT ACHVGGGTWTCTAAT-3′). Data were imported into QIIME2[100] using standard methods and the developer's docker container (qiime2/core:2018.8). V4 region amplicons were assembled and denoised using the QIIME2 method 'DADA2 denoise-paired'. Processed V4 amplicons were grouped into amplicon sequence variants (ASVs) with 100% sequence similarity. Two reagent-only samples per plate were included to indicate the potential degree of contamination in the absence of bacteria in the samples (three 96-well plates used in total). All six reagent-only samples produced very low assembled 16 S read counts of between 74 to 372 total reads (average 149 reads), indicating neg-ligible contamination of the actual samples which had between 11,929 and 108,169 total reads (average 42,335 reads). ASVs were classified using a pre-trained classifier based on SILVA (release 132)[101], a com-prehensive database that provides accurate annotations[102]. ASV counts per sample were exported as biom files from a QIIME2 artifact and converted into a human readable tsv file using "biom convert". Read counts per sample were rarefied to 11,929 reads per sample (the lowest count among the 242 samples) using the "rrarefy" command in the R package "vegan" (version 2.6-4), and normalized read counts were calculated per sample by dividing the number of reads associated with each ASV by the total number of reads assigned across ASVs. Taxo-nomic identifications used are directly provided by SILVA (release 132)[101]. Raw 16 S rRNA can be downloaded from public database (SRA BioProject PRJNA911205; All sample accessions are available in Sup-plementary Data 2).

## Whole Metagenome Shotgun (WMS) sequencing and GM profiling

For the WMS sequence analysis, samples were divided into "high" and "low" SD and PS based on the distribution of these values across the sample set. Samples above the average value + 0.5 standard deviations were considered "high" and samples below the average value −0.5 standard deviations were considered "low" (35 "low-SD", 43 "high-SD, 36 "low-PS" and 32 "high-PS"; Fig. 1c). After this selection, whole Metagenome Shotgun (WMS) datasets for were generated for 178 of the 242 samples (89 samples from mothers and 89 from their respective children) on the Illumina NovaSeq S4 (150 bp paired end reads). WMS sequencing was performed by the Genome Technology Access Center at McDonnell Genome Institute (GTAC@MGI) at Washington University in St. Louis School of Medicine, USA. For each sample, ~6Gbp was generated, producing between 2.7 and 237.2 mil-lion reads per sample (average 61.5 million reads). The reads for all 178 samples (89 mothers and 89 children) were cleaned of barcodes, adapters, and low-quality ends using Trimmomatic[103] (version 0.36). The BMTagger program (installed using conda, version 3.101) was used to identify human contaminant reads using the human reference genome (GRCh38.98[104]). Reads identified as human were removed to produce final paired-end fastq per each of the 178 samples. Raw WMS read data can be downloaded from public database (SRA BioProject

PRJNA911205; All sample accessions are available in Supplementary Data 2).

The 178 WMS samples were mapped against the Unified Human Gastrointestinal Genome (UHGG) collection, comprising 204,938 nonredundant genomes from 4,644 gut prokaryotes, each theoretically representing an individual bacterial or archaeal species (95% average nucleotide identity[42]) using bowtie2[48] (v2.3.5.1). The profile module of the inStrain[105] program (v1.0.0) was then run to generate sequencing breadth and depth of coverage statistics for every genome, in addition to nucleotide diversity measures per genome per sample. The depth of coverage values were normalized within every sample by dividing each genome's depth by the sum of the depths across all genomes.

The 178 WMS samples were also used as input for HUMAnN3[46,47] (version 3), which was run from the biobakery/humann docker container (latest version as of October 2020) using the Chocophlan nucleotide database and Uniref90[106] protein database. HUMAnN3 runs the MetaPhlAn (version 3)[47] program as an intermediate step to assign organism-specific functional profiling, and the developer-provided Metaphlan3[47] bowtie2[48] database was used for this intermediate step. The HUMAnN3 pipeline was used to generate MetaCyc[45] pathway abundance per sample. The "humann_renorm_table" script (included in the HUMAnN3 distribution) was used to convert Reads Per Kilobase (RPK) values in the MetaCyc abundance table to a normalized value, Copies Per Million (CPM), which can be compared across samples.

Cytoscape (v 3.10.0) was used to construct networks (Fig. 8) connecting high-SD associated genomes (from Figs. 6 and 7) to high-SD associated pathways (Table 2), with node sizes indicating the mean decrease in accuracy (MDA) values from the random forest analysis, and edge thickness and darkness indicate the proportion of reads that each genome contributes to the pathways (as calculated from the HUMAnN3 output).

## Statistics and reproducibility

No statistical method was used to predetermine sample size, with sample sizes selected. Investigators performing sample collection (as previously described for the Early Life Adversity Biological Embedding and Risk for Developmental Precursors of Mental Disorders Study (eLABE; details provided in ref. 30)). Investigators performing sample collection were blinded, but investigators performing bioinformatic analyses were not blinded to the sample metadata out of necessity. Randomization of sequenced samples was not performed. No eligible datasets meeting criteria for inclusion were excluded from the analyses. Of the available and eligible 134 mother:child pairs of samples, 121 were selected for 16 S sequencing analysis, and of those, 89 that were either at the high or the low range of either SD or PS were selected for whole metagenome shotgun sequencing, in order to facilitate high-vs-low statistical comparisons.

## Detailed statistical analysis

For the 16 S rRNA/ASV sample analysis, Faith phylogenetic diversity values[35] (for α-diversity) and UniFrac[38] weighted and unweighted distance, as well as Aitchison distance[37] (for β-diversity) were calculated for each sample using QIIME2[107]. Also for β-diversity, Bray-Curtis distance diversity values were calculated using the "vegdist" function in the "vegan" R library (version 2.6-4).

Significant differences of metadata classifications between Low-SD vs High-SD and Low-PS vs High-PS sample sets were quantified using several different approaches (Table 1, Supplementary Data 1a). The correlation between SD and PS was tested using a T-statistic test of the Pearson correlation[108] (all Pearson correlation statistical results presented in Supplementary Data 1b). The Shapiro-Wilk test for normality was used to determine whether each set of metadata followed a normal distribution prior to performing comparisons of differences of means[109] (all Shapiro-Wilk test results and determinations of normality

are provided in Supplementary Data 1c). For data determined to be normally distributed (e.g., mother's delivery age, healthy eating index, mother's third trimester TNF-α levels, child birthweight), two-sided T-tests with unequal variance[110] were performed (all T-test statistics and significance values are provided in Supplementary Data 1d). For data that were determined not to be normally distributed using the Shapiro-Wilk test, Mann–Whitney U-tests[111] (also known as Wilcoxon Rank Sum tests) were used to determine significant differences of mean values (all Mann–Whitney U-test statistics and significance values are provided in Supplementary Data 1e). Significant differences in categorical variables (i.e., race, sex of child, route of delivery, high-vs-low breast milk feeding) were performed using the Chi-Square test[112] (all Chi-Square test statistics and significance values are provided in Supplementary Data 1f).

For the differences in β-diversity between and within sample groups (Fig. 4 and Supplementary Fig. 3), Benjamini-Hochberg false discovery rate (FDR) correction was performed to correct for multiple testing[113]. FDR correction was not performed for the comparisons of sample metadata presented in Table 1 and Supplementary Data 1a because these were tests performed for an overview of associated metadata prior to testing differences in the GM and were not considered as part of the results. Significant differences in components of SD and PS were expected since samples were chosen from the extremes of phenotype for SD and PS to improve detection of GM differences between groups[30].

ASV-based sample clustering was performed using non-metric multidimensional scaling (NMDS)[114] using the "metaMDS" in the 'vegan' R package (v 2.6-4), with setting k = 2. Dirichlet Multinomial Mixtures (DMM)[39] clustering was performed to separate samples into cluster groups independently of the NMDS plot, using the 'DirichletMultinomial' package (v 1.40.0) in R, using ASVs detected in at least 3 samples. Clustering was performed and tested from n = 1 to n = 5, and an optimal cluster number per comparison was selected based on minimum log posterior loss correction (lplc) values from DMM[39], as well as using silhouette scores[40] ("cluster" R package, version 2.1.4) and prediction strength values[41] ("flexible procedures for clustering; fpc" R package, version 2.2-10), using the NMDS distance matrix and DMM cluster assignments as input. Cluster strength results are provided in Supplementary Data 1g.

For the WMS sequence analysis, samples were divided into "high" and "low" SD and PS based on the distribution of these values across the sample set. Samples above the average value + 0.5 standard deviations were considered "high" and samples below the average value −0.5 standard deviations were considered "low" (35 "low-SD", 43 "high-SD", 36 "low-PS" and 32 "high-PS"; Fig. 1c). The same approach was used to separate samples into "high" and "low" sample sets based on inflammatory marker data (IL-6, IL-8, IL-10, and TNFα).

To identify bacterial taxa that strongly predict mothers' SD and PS scores, we analyzed taxonomic and pathway GM profiles using three approaches. First, a supervised machine-learning approach (Random Forest[49]) that identifies non-linear relationships from high dimensional and dependent data[49] was used to (i) quantify the ability to predict metadata classification based on the microbiome profiles, indicative of the overall association between the microbiome and the composite scores, and (ii) for each comparison, identify the specific genomes or pathways that most strongly differentiate between high and low SD and PS scores, ranked based on "mean decrease in accuracy" (MDA; representing the percentage of prediction accuracy that would be lost if a genome/pathway was excluded from the RF training). When discussing RF results, "predictors" is used to indicate the genomes or pathways used in the final RF model (minimum MDA 0.1%). RF was run using the "randomForest" package in R (v4.7-11). The generalization error of the model was evaluated by out of bag (OOB) error. MDA values for RF are shown in Supplementary Data 1i for every genome and Supplementary Data 1j for every pathway for every comparison in

both the mothers and the children. The association of the metadata with the microbiome was quantified using the RF classification accuracy, and the significance of the accuracy was measured using binomial distribution tests[115] (Fig. 5a) with FDR correction applied to correct for the number of tests[113] (all binomial distribution test statistics and significance values are provided in Supplementary Data 1h, and account for differences in sample numbers in each "high" vs "low" comparison). RF model accuracy was also examined using receiver operating characteristic (ROC) curves, quantified using the area under the curve (AUC) (Fig. 5b) generated using the R library "ROCR" (version 1.0-11). Significance values for the ROC curves were assigned by Two-sided Mann–Whitney U statistics[116], using the "roc.area" function in the "verification" R package, version 1.42.

Second, linear discriminant analysis effect size (LEfSe[50], Galaxy Version 1.0), the most frequently used statistical tool to determine significant differences in microbiome member abundance[117], was used for differential genome abundance testing (default settings at $P \leq 0.05$ for significance) for the non-parametric factorial Kruskal-Wallis (KW) sum-rank test, and requiring a linear discriminant analysis (LDA) ES (effect size) of at least 2 in order to identify differentially abundant taxa. The same approach was used for the pathway analysis, but the ES test cutoff applied was reduced to a value of 1 instead of 2, since the ES is designed for the more sparse nature of metagenomic abundance data[50]. However, the same RF MDA and LEfSe Kruskal-Wallis cutoffs were applied for pathway analysis. False Discovery Rate (FDR) correction was performed for the LEfSe KW significance values, among the genomes and pathways in the top 25 genomes / pathways identified by the RF analysis (which were the only ones considered for discussion purposes). LEfSe LDA values effect size values and KW $P$-values (uncorrected and FDR-corrected) are shown in Supplementary Data 1i for every genome, and Supplementary Data 1j for every pathway, for every comparison in both the mothers and the children.

Third, ANCOM-BC2[51], a differential abundance tool for microbiome data which estimates the unknown sampling fractions and corrects the bias induced by their differences among samples, was ran using the "ANCOMBC" R package (version 2.1.4), using genomes detected in at least 3 samples, and the setting prv_cut = 0.05. This was performed to provide additional confidence in results identified by Random Forest and LEfSe, but was not applied as a filter for identifying taxa of interest to present in figures. FDR-corrected $P$-values both against all genomes and against the top 25 genomes identified by RF in each comparison, are provided in Supplementary Data 1i.

All programs and code used in the analysis are described above. All samples tested represent distinct biological samples, with no technical replicates sequenced or used for any statistical analysis. All relevant metadata required to reproduce the reported analyses are provided in Supplementary Data 2. The study complies with the "Strengthening The Organization and Reporting of Microbiome Studies" (STORMS) criteria.

### Reporting summary
Further information on research design is available in the Nature Portfolio Reporting Summary linked to this article.

## Data availability
The raw 16 S sequence reads and whole metagenome shotgun (WMS) sequence reads generated in this study have been deposited in the NCBI Sequence Read Archive (SRA) database under BioProject number PRJNA911205. 16 S taxonomic identifications were produced using the SILVA database (release 132). WMS sequences were mapped to the Unified Human Gastrointestinal Genome (UHGG) database (version 1), and pathways were quantified from mapping results using HUMAnN3 (version 3), which is based on UniProt/UniRef 2019_01 sequences and annotations. The detailed sample metadata matched to SRA accession numbers, normalized abundance data for all 16 S ASVs, WMS genomes

and WMS pathways for every sample data are available at in Supplementary Data 2. ASV sequences are available in Supplementary Data 3. Individuals have been de-identified using random identifiers. Individuals have been de-identified using random identifiers. Results from all statistical comparisons are provided in Supplementary Data 1.

## Code availability
Custom R scripts, example input files and instructions for usage for all analyses are available at Protocol Exchange (https://doi.org/10.21203/rs.3.pex-2283/v1)[118].

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

## Acknowledgements

This work was supported in part by NIMH grant RO1 MH113883, (BBW, BAR, IMN, PPT, PJM, SKE, JLL, CER, TAS, CDS, DMB, GEM, EC, JM, MM) March of Dimes (SKE), Children's Discovery Institute II MD-II-2015-489 (BBW, IMN), Biobank Core P30DK052574 (PIT), REDCap database with Clinical and Translational Science Award (CTSA) Grant [UL1 TR000448] and Siteman Comprehensive Cancer Center and NCI Cancer Center Support Grant P30 CA091842. We are thankful to the families involved in the study, the MARCH of Dime Premature Research Center at Washington University and to members of the Washington University in St. Louis' GTAC@MGI for technical support related to library construction and sequencing.

## Author contributions

Conceived and designed the study: B.B.W., M.M., J.L.L., D.M.B., C.E.R., C.D.S. Assembled the cohort, collected specimens: I.M.N., L.L., S.K.E., T.A.S. Biological and clinical data database maintenance: I.M.N., R.E.B., J.D.W., C.H.-M. Sample preparation and extraction of DNA: I.M.N., R.E.B., J.D.W., C.H.-M. Performed data analysis: B.A.R., M.M., J.M., J.P.M., G.E.M., E.C. Interpreted the data: B.A.R., M.M., B.B.W., J.M., P.I.T., I.M.N. Wrote the paper: B.A.R., M.M., B.B.W. All authors approve the manuscript for publication.

## Competing interests

The authors declare no competing interests.

## Additional information

**Barbara B. Warner** [1,12] ✉, **Bruce A. Rosa** [2,12], **I. Malick Ndao**[1], **Phillip I. Tarr** [1,3], **J. Philip Miller** [4], **Sarah K. England** [5], **Joan L. Luby**[6], **Cynthia E. Rogers** [7], **Carla Hall-Moore**[1], **Renay E. Bryant**[1], **Jacqueline D. Wang**[1], **Laura A. Linneman**[1], **Tara A. Smyser** [6], **Christopher D. Smyser** [8], **Deanna M. Barch** [9], **Gregory E. Miller** [10], **Edith Chen**[10], **John Martin**[2] & **Makedonka Mitreva** [11] ✉

[1]Department of Pediatrics, Washington University School of Medicine in St. Louis, St. Louis, MO 63110, USA. [2]Department of Medicine, Washington University School of Medicine in St. Louis, St. Louis, MO 63110, USA. [3]Department of Molecular Microbiology, Washington University School of Medicine in St. Louis, St. Louis, MO 63110, USA. [4]Institute for Informatics, Data Science and Biostatistics, Washington University School of Medicine in St. Louis, St. Louis, MO 63110, USA. [5]Department of Obstetrics and Gynecology, Center for Reproductive Health Sciences, Washington University School of Medicine in St. Louis, St. Louis, MO 63110, USA. [6]Department of Psychiatry, Washington University School of Medicine in St. Louis, St. Louis, MO 63110, USA. [7]Departments of Psychiatry and Pediatrics, Washington University School of Medicine in St. Louis, St. Louis, MO 63110, USA. [8]Departments of Neurology, Pediatrics and Radiology, Washington University School of Medicine in St. Louis, St. Louis, MO 63110, USA. [9]Department of Psychological and Brain Sciences, Psychiatry, & Radiology, Washington University in St. Louis, St. Louis, MO 63130, USA. [10]Institute for Policy Research & Department of Psychology, Northwestern University, Evanston, IL 60208, USA. [11]Departments of Medicine and Genetics, and McDonnell Genome Institute, Washington University School of Medicine in St. Louis, St. Louis, MO 63110, USA. [12]These author contributed equally: Barbara B. Warner, Bruce A. Rosa. ✉e-mail: warnerbb@wustl.edu; mmitreva@wustl.edu

