## [Peer Review File · Nature Communications]

REVIEWER COMMENTS

Reviewer #1 (Remarks to the Author):

This paper investigates the gut microbiome of mother-child dyads as a potential mechanism that can explain how social disadvantages and psychological stressors may lead to health disparities. The authors investigate both compositional and functional characteristics of the maternal and child microbiome as well as prenatal maternal cytokine concentrations. The manuscript presents interesting results in a relevant area of research where there is yet a scarcity of studies. However, there are still some major analytical issues that need to be attended to before the paper is ready for publication.

MAJOR ISSUES

1) P5 | 114. $R=0.395$ is not considered 'highly correlated'. Do the authors mean 'not highly correlated'? Then what follows would make sense.

2) P9 | 196. The authors describe using a "two-round approach", which basically means using the data twice, once for feature selection and then again with the selected features. This strategy causes overfitting (see also the longer point further below about the RF description). As an apparent justification for this approach, paper (39) is cited. However, this is neither a methods paper, nor a paper where there is any particular focus on a "two round approach". In fact, the cited paper describes a procedure for overfitting: using the data twice (feature selection and final model fitting) and furthermore, arbitrarily changing default parameters without any justification (this must either be derived by hyperparameter tuning or, alternatively, changing the default must be otherwise justified).

3) P 23 | 503. Continuous variables are categorized into high/low. What is the motivation to categorize the data as compared to modeling it as continuous? By disregarding the data points that lie in the average SD/PS range, the authors disregard information without any justification. Do the authors believe that the relation between SD or PS and the microbiota is different for people who have average SD/PS or that it is not linear along the continuum of SD/PS? Did the authors investigate the same question using all data points of SD/PS instead as well? All analyses could also have been used with SD/PS as continuous scores. Finally, did the authors pre-register that they will split the data based on ± 0.5 SD beforehand?

4) p 23 | 503. The authors solely used Shannon index as a measure of alpha diversity and Bray Curtis similarity as a measure of beta diversity. It is advisable to perform the analyses on a variety of alpha/beta diversity measures in order to test how robust findings are when other metrics are used and also to emphasize different aspects of alpha/beta diversity (see Bastiaansen et al. 2022 for a good guide on this topic and why this is important). Please add at least one alpha diversity measure that includes phylogenetic information, such as Faith index. Furthermore, for beta diversity, please also perform the analyses for a phylogenetic beta diversity score (e.g., UniFrac or weighted UniFrac), as well as using Aitchison distance (please be aware of Gloor et al. 2017, as no CoDa tools were utilized). Adding these additional measures will make it possible to report if the results are robust, or very much depend on the metric used. The additional analyses can be placed in a supplement, and in the Results, the information derived from these extra analyses can be used to report about the robustness of the findings.

5) P24 | 525. "Multivariate testing considering significant covariates..." I think there are two important points to be addressed in this section:

a) As I will explain, using statistical significance to select covariates is not a valid approach. If one attempts to understand the relation between SD/PS and the gut microbiota, choosing covariates based on statistical significance can bias results. P-values were not designed for this purpose and say nothing about the nature of the relations between variables. For example, a collider (see Wysocki et al. 2022 or Cinelli et al. 2020) would be statistically correlated to predictor and outcome and would induce collider bias if included. A mediator also would be correlated to predictor and outcome and would bias the total effect estimate. Lastly, a variable may be related to predictor and outcome in the expected direction but not pass the threshold of significance; this does not mean it cannot be an important confounder (blocker).

To select covariates, it is advisable to propose a theoretical model, which in turn guides the analysis (see Wysocki et al., 2022). For this reason, I would recommend the authors make clear assumptions like they for example make in a later sentence in this paragraph: "GA for this cohort was restricted to term (≥ 37 weeks), with resulting GA and birthweight differences without precedent for GM differences in the range of values. Maternal age similarly does not have precedence within these age distributions for GM differences beyond that controlled for in SD". Once the authors have described their theoretical model, e.g., in the form of a directed acyclic graph (Wysocki et al. 2022; Cinelli et al. 2020), they can evaluate if their data are compatible with their model(s). Alternatively, the SEM framework allows evaluating several candidate models based on model fit criteria.

Note that if the authors' goal is not to understand the relations but solely to make good predictions, then covariate selection based on statistical significance is still not a good approach. To maximize predictive accuracy, model selection based on information criteria or out of sample prediction performance (e.g., using cross validation) would make sense then. Alternatively, regularized models with broad covariate selection can also make sense for prediction accuracy.

b) The authors write "...not performed on covariates within the SEM model...". Looking at the SEM model in the cited publication, it appears that the covariates the authors seem to refer to are either only reflected in variation in SD or PS. Furthermore, those variables will only load partially onto the

latent variables. The remaining variance of each predictor would be ignored. So, the assumption that “correcting for it would eliminate the goal of the analysis.” seems questionable to me, especially since the authors perform separate analyses for SD and PS. Nevertheless, I appreciate that the authors state this and several other assumptions and the critical reader is made aware of it. If they could somehow explain/derive that this last assumption can actually be made, that would make the ground of the analyses appear much stronger. Now, it seems shaky to me.

6) P24 | 543. Significant differences in SD and PS between clusters were tested. Did the authors evaluate absolute measures of cluster strength such as Prediction Strength and Silhouette Index? If those aren't high enough (e.g., ≥ 0.9 and 0.5 , respectively), it is unlikely that these clusters will be reproducible also in other data (see Koren et al. 2013). I'm assuming that the authors are interested in ensuring reproducibility of clusters beyond their data set. But if this is not the case, then the reasons for this should be explained in the manuscript. If, as expected, they would like to ensure reproducibility, the paper by Koren can provide good guidance.

7) P25 | 560. The description of the Random Forest algorithm is insufficient. How did the authors make sure that overfitting was avoided? The descriptions says “generalization error of the model was evaluated by out of bag (OOB) error”. The OOB error is a valid measure of accuracy. However, the authors describe that they selected features before calculating final accuracy. Did they split the data into training and testing data or did they use cross validation to do so? If so, what are the specifics of that ($p = ? / k = ?$). If the authors ran a model on the full data to calculate the importance scores and then reran the model again on the same data only using the most important variables (as described in the paper they cited), this will cause overfitting of the data and all the presented RF accuracy scores are an overestimation of the true accuracy. If this is what the authors did, please report the accuracy of the model that is not overfit. Also, please indicate whether you changed the hyperparameters. It is important to avoid overfitting when tuning models as well. Always make sure that the data you use to evaluate accuracy (here OOB/ROC), has never been “seen” by the model, neither to select features, nor to tune parameters. This is a very common mistake in our field. You may report OOB of the model that has not been tuned (no hyperparameter tuning or feature selection) or you may report accuracy after any kind of tuning, using a test data set that has not been used for tuning/feature selection.

8) P 10 | 211. Are the authors sure that this is how MDA was calculated? Judging from the R packages they used it is more likely that the genome/pathway was not removed but permuted. In case permutation was indeed used, please modify the description.

9) p 26 | 583. The authors used LefSe for differential abundance testing. A recent Nature Communications benchmark study (Nearing et al. 2022) was careful to make any clear recommendations but noted that the field may want to avoid LefSe. Despite this benchmark study, I do not want to ask the authors to entirely replace their LefSe analysis if they have valid arguments to

choose this method over other, in my opinion better, methods. I do however ask the authors in any case to add at least one of the CoDa methods such as ANCOM (II/BC) or ALDEX2 (again, see Gloor et al., 2017 for an explanation of CoDa methods). The results of this additional method can be added to the existing table that shows RF and LEfSe results or otherwise it should be noted somewhere that the results were (or were not) robust depending on the methods used. The reason I ask this is that the field has not yet established a state of the art method that performs best in most scenarios. Therefore, and in order to avoid a reproducibility crisis, we must ensure that findings are robust, independent of method choice.

MINOR ISSUES

10) Table 1: refers to 'patients'. Should be participants. Also: please explain all abbreviations (i.e. MGS, Psych) in the notes of this and other tables.

11) Fig.1. Legend: clarify what WMS is.

12) P6 | 134. I find this explanation confusing. Alpha diversity between two samples can be exactly the same and yet the samples can be very dissimilar. The text reads as if alpha diversity is responsible for the result of beta-diversity, while these statistics just reflect different parts of the same snapshot and cannot cause each other. A possible alternative interpretation would be that the high-SD children's GM is more adult-like compared to low-SD children as reflected in higher similarity to their mother's GM and higher alpha diversity.

13) p 7 | 146. What led the authors to state that low-SD GMs are presumably healthier? How would they quantify a healthy GM and how did they investigate that the low-SD GMs are more similar to that healthy GM that they have in mind than high-SD GMs? Without any quantification, this statement is not evidence-based and can be misleading. While it may be tempting to point towards what might seem obvious, it is still necessary to do that based on data before making such an interpretation. In the Conclusion section, the interpretation of this finding is much more valid.

14) p 15 | 340. "...although greater SD is associated with greater maternal IL-6 in the full eLAGE cohort." Please add a reference for this.

15) p 16 | line 351. "...the impact of exposure to both SD and PS on GM structure and function for mothers and their infants." The study is observational in nature; therefore, causal language (i.e. impact) should be avoided when interpreting results.

16) p 27 | 598. Please consider making your code publicly available such that reviewers can ensure correctness, as well as for other researchers to learn and reproduce your analysis.

References

Bastiaanssen, Thomaz F. S., Thomas P. Quinn, and Amy Loughman. 2022. "Treating Bugs as Features: A Compositional Guide to the Statistical Analysis of the Microbiome-Gut-Brain Axis." arXiv. <https://arxiv.org/abs/2207.12475>.

Cinelli, Carlos, Andrew Forney, and Judea Pearl. 2020. "A Crash Course in Good and Bad Controls." SSRN Electronic Journal. <https://doi.org/10.2139/ssrn.3689437>.

Gloor, Gregory B., Jean M. Macklaim, Vera Pawlowsky-Glahn, and Juan J. Egozcue. 2017. "Microbiome Datasets Are Compositional: And This Is Not Optional." *Frontiers in Microbiology* 8 (November). <https://doi.org/10.3389/fmicb.2017.02224>.

Koren, Omry, Dan Knights, Antonio Gonzalez, Levi Waldron, Nicola Segata, Rob Knight, Curtis Huttenhower, and Ruth E. Ley. 2013. "A Guide to Enterotypes Across the Human Body: Meta-Analysis of Microbial Community Structures in Human Microbiome Datasets." Edited by Jonathan A. Eisen. *PLoS Computational Biology* 9 (1): e1002863. <https://doi.org/10.1371/journal.pcbi.1002863>.

Nearing, Jacob T., Gavin M. Douglas, Molly G. Hayes, Jocelyn MacDonald, Dhvani K. Desai, Nicole Allward, Casey M. A. Jones, et al. 2022. "Microbiome Differential Abundance Methods Produce Different Results Across 38 Datasets." *Nature Communications* 13 (1): 342. <https://doi.org/10.1038/s41467-022-28034-z>.

Wysocki, Anna C., Katherine M. Lawson, and Mijke Rhemtulla. 2022. "Statistical Control Requires Causal Justification." *Advances in Methods and Practices in Psychological Science* 5 (2). <https://doi.org/10.1177/25152459221095823>.

Reviewer #2 (Remarks to the Author):

This reviewer wished to congratulate the authors on an exceptionally well written manuscript. In particular, the methods section and description of statistical analyses are extremely detailed and wholly transparent. The inclusion of substantial supplementary material detailing all test results and associated sample metadata is commendable. Links with SRA accession numbers will facilitate replication findings with ease.

The authors refer to plasticity of GM and potential to intervene as an important route to impact for this study. Is the microbiota the driving force behind DS/PS though? Or are environmental factors/exposures associated with DS/PS responsible for shaping GM? Will modulation of GM yield social & psycholological equity? This feels like a reach?

The finding that SD/PS is negatively associated with breastfeeding – which in turn is linked with GM structure/diversity is one of great interest/importance and may inform future practice to understand barriers to breastfeeding in high SD/PS mother-infant dyads as well as promoting breastfeeding within these populations.

The authors show that there may be age-dependent dysbiotic mechanisms implicated in high DS/PS states. Predictive features of DS/PS are contrasting between mothers and infants. They illustrate that in infants this dysbiosis may be associated with fewer gut anaerobes such as *Bifidobacterium infantis*, *Veillonella parvula* and *Colinsella* spp. These results tell a lovely story because they fit well with the findings of reduced breast milk feeding in high SD/PS infants and were a joy to read. I would like to see more made of the metabolic pathways though. If the authors could link metabolic pathways with these strict anaerobes which appear to be predictive of low SD/PS that would represent a step toward identifying ways in which they might achieve the promises alluded to in the introduction: namely identifying ways to modulate the GM to alleviate impacts of high SD/PS.

Could the authors validate their predictive factors identified by RF by employing other classification methods such as logistic regression models using only the discriminative features identified?

Abstract

Line 62 – “compare cohort” should read “compare cohorts”

Line 65-66 – Can the authors provide summary values to quantify these statements? How log is inter-individual GM similarity in high-DS/high-PS mothers? How does this value compare to low-DS/PS mothers? Is this difference significant?

Line 67 – What is the predictive accuracy of IL-6?

Introduction

A very well written and thoughtful introduction to the subject matter. Kudos to the authors!

Line 88 – “interaction” should be interaction?

Line 89-90 – Could the authors cite some literature describing the importance of gut microbial plasticity and it's therapeutic potential? Suggest something like Marchesi et al., 2016 (<https://doi.org/10.1136/gutjnl-2015-309990>)?

Results

The authors present a vast amount of data in a clear and concise way - congratulations again. The subheadings could be more impactful though. Instead of describing the comparison made, could these highlight the most important findings of those comparisons?

No stratification is explained for the RF models. How many/which samples were used as the training and test datasets? This is extremely important to understand the validity of the model.

Multiple figure panels are not referenced in text. This is especially striking for figure 7B. Why are child IL-6 links with microbiota not discussed in text? Especially as the association of multiple *Bifidobacterium* and *Colinsella* spp. (which were differential between high/low SD/PS infants) are associated with these stratified groups. It feels as though the results are building up toward something really special but the authors don't really nail selling the big findings at the end??

Line-by-line comments included below:

Line 104 – Is the eLABE study protocol published? Could the authors link out to this here?

Line 105-6 – Assume this means recruitment occurred retrospectively? Could the authors make this clear in both line 102 and the abstract?

Line 106-7 – “SD and PS status had significant variability” is this between high and low groups? This is not made clear in text. Authors' use of the word “significant” implies some sort of statistical testing was performed but no reporting of results. Please specify what the comparison is by which significance is identified or use alternative language such as “substantial”.

Line 112 – Appreciate this is a contentious opinion but this reviewer does not consider 16S rRNA gene sequencing to be metagenomics. Consider instead “we performed targeted bacterial 16S rRNA gene sequencing”.

Line 130 – Consider adding “significantly” positively correlated with SD and PS?

Line 132 – Would be useful to reiterate the significance of the difference between high SD/PS mothers and breastfeeding frequency here to really drive this point home. This is a really nice, important finding!!

Line 139-43 – Another nice finding and fits well to show that SD/PS have a greater impact on GM in earlier than later life. Do the children clustering in the lowest SD/PS scores cluster with their mothers if the authors combined all mothers/children together though? This would work really nicely to illustrate the point that >SD/PS = <BM = >GM diversity among children = >Similarity to maternal GM and would tie the figure together really nicely.

Line 144 – This is confusing. Do you have longitudinal data from participants to compare longitudinal GM profiles? Surely this is the only way to show consistent GM profiles among healthier, or low SD/PS individuals? Is this data not just finding that high SD/PS mothers are more similar to one another. This would suggest some kind of environmental pressure has a homogenising effect on shaping GMs?

Line 144-57 – B-diversity here is reported as Bray-Curtis similarity!?. Nowhere in methods is a conversion mentioned (assuming 1 – BC-dissim value??). If this is the case please state. Otherwise please consider these results may be incorrectly inferred? Is there actually >dissimilarity between low SD/PS mothers rather than >similarity?? This section may need to be revised but the concept is very interesting. Whether there is a homogenising or individualising influence of SD/PS raises questions about what the environmental factors are that promote these.

Line 168 – Only 11 bacterial genomes frequently detected are listed in text!?

Line 169-71 – Perhaps reference supplementary figure 3 here?

Line 197 – Please reference individual panels of figure 4 that illustrate the points you are making. This comment applies to most of this section of the MS.

Line 208-11 – This explanation feels more like methods. Could this text be moved there and the results section here focus more on describing the bacterial features associated with model accuracy?

Line 226 – “Bifidobacteria” should not be italicised.

Line 274-6 – Might this suggest age-specific GMs associated with >SD/PS? Are there any pathways that are imbalances between the SD/PS groups across mothers/infants? Could you perform some kind of network analysis to show which pathways are regularly associated with each bacterial genome and how these are implicated in SD/PS grouping?

Line 322-5 – Again, need to know how many samples used to train / test RF models. Were models built using only cytokines? Could models built using cytokine, functional and taxonomic profiles provide better classification?

Line 340 – Seems strange not to reference the full eLABE cohort or provide values to quantify this claim re: IL-6 here?

Conclusions

The authors deserve substantial credit for attempting to provide a balanced summary of their data. They explain the important findings but also highlight the limitations of association studies. It is this reviewer's personal opinion that the authors could provide slightly more context to the breast-milk feeding/anaerobic taxa/high IL-6 links in high SD/PS groups though. The authors are almost too cautious with a very exciting dataset.

Line 350 – Is this a prospectively assembled cohort? If the selection criteria include reaching 4 months corrected age and are subsampled from the larger eLABE cohort surely recruitment is retrospective??

Methods

Line 459 - Please be consistent with capitalisation of Qiime2 throughout

Line 475 – Please explain 178 samples used for WMS. 35 low-SD + 43 high-SD + 36 low-PS + 32 high-PS = 146 samples. If only 89 samples from this cohort of 146 were chosen for WMS how were these samples selected?

Line 492 – Was HUMAnN2 or HUMAnN3 used? Tool name is missing – text suggests HUMAnN3 but citation suggests HUMAnN2. Please correct citation if necessary (<https://doi.org/10.7554/eLife.65088>)

Line 545 – Is hclust the best way to do this? Are 4 clusters per mothers/children the most optimal to describe the communities? Would these numbers be consistent if using DMM and confirming with lowest Laplace approximation score as in Beck et al., 2022, (<https://doi.org/10.1038/s41564-022-01213-w>)?

Blue = author response

Green = changes made in the MS (bold green font used to highlight the change that was made)

REVIEWER COMMENTS

Reviewer #1 (Remarks to the Author):

This paper investigates the gut microbiome of mother-child dyads as a potential mechanism that can explain how social disadvantages and psychological stressors may lead to health disparities. The authors investigate both compositional and functional characteristics of the maternal and child microbiome as well as prenatal maternal cytokine concentrations. The manuscript presents interesting results in a relevant area of research where there is yet a scarcity of studies. However, there are still some major analytical issues that need to be attended to before the paper is ready for publication.

MAJOR ISSUES

R1-1: P5 | 114. $R=0.395$ is not considered 'highly correlated'. Do the authors mean 'not highly correlated'? Then what follows would make sense.

Author response: This has been revised to read "significantly but only moderately correlated" (lines 124-125).

R1-2: P9 | 196. The authors describe using a "two-round approach", which basically means using the data twice, once for feature selection and then again with the selected features. This strategy causes overfitting (see also the longer point further below about the RF description). As an apparent justification for this approach, paper (39) is cited. However, this is neither a methods paper, nor a paper where there is any particular focus on a "two round approach". In fact, the cited paper describes a procedure for overfitting: using the data twice (feature selection and final model fitting) and furthermore, arbitrarily changing default parameters without any justification (this must either be derived by hyperparameter tuning or, alternatively, changing the default must be otherwise justified).

Author response: After re-considering the approach and the paper we had used as a guidance for our analysis, we do agree with the reviewer that the "two-round approach" is overfitting the data and that statistically it is more reasonable to present the accuracy from the first "round" only. Therefore, now we have provided the result from only the first-round data, which still provides strong accuracy and P values ($P < 0.01$ after FDR correction) for the SD and PS scores. This request resulted in replacing Figures 5, 6 and 7, so that all accuracy values and MDA values were updated to reflect a single out-of-bag (OOB) RF run. References to the two-round approach have been removed throughout the text, figures and supplementary figures and tables.

R1-3: P 23 | 503. Continuous variables are categorized into high/low. What is the motivation to categorize the data as compared to modeling it as continuous? By disregarding the data points that lie in the average SD/PS range, the authors disregard information without any justification. Do the authors believe that the relation between SD or PS and the microbiota is different for people who have average SD/PS or that it is not linear along the continuum of SD/PS? Did the authors investigate the same question using all data points of SD/PS instead as well? All analyses could also have been used with SD/PS as continuous scores. Finally, did the authors pre-register that they will split the data based on ± 0.5 SD beforehand?

Author response: As described in the methods and in Luby et al 2023 (PMID: 36914799), SD and PS are highly complex variables that combine many components of metadata including self-reported data, which is likely to introduce noise. Dividing the dataset into high vs low substantially reduces the impact of this expected noise, by ensuring that the categorization of each sample into "high" and "low" is as confident as possible.

The authors did not pre-register this split data decision, since the samples were collected as part of a larger overall study with additional research goals (see Luby et al, 2023). However, the plan was always to undertake a stepwise approach, (i) targeted 16S rRNA gene sequencing on all samples, followed by (ii) whole metagenome shotgun sequencing on samples selected with the intention of comparing high vs low SD/PS scores. We believe that starting with comparing the extremes of the spectrum and following up in the future

with a larger cohort that will allow comprehensive comparisons and undertaking a continuous linear analysis is the way to approach complicated metadata with complex variables and whole metagenomic shotgun data type.

R1-4: p 23 | 503. The authors solely used Shannon index as a measure of alpha diversity and Bray Curtis similarity as a measure of beta diversity. It is advisable to perform the analyses on a variety of alpha/beta diversity measures in order to test how robust findings are when other metrics are used and also to emphasize different aspects of alpha/beta diversity (see Bastiaansen et al. 2022 for a good guide on this topic and why this is important). Please add at least one alpha diversity measure that includes phylogenetic information, such as Faith index. Furthermore, for beta diversity, please also perform the analyses for a phylogenetic beta diversity score (e.g., UniFrac or weighted UniFrac), as well as using Aitchison distance (please be aware of Gloor et al. 2017, as no CoDa tools were utilized). Adding these additional measures will make it possible to report if the results are robust, or very much depend on the metric used. The additional analyses can be placed in a supplement, and in the Results, the information derived from these extra analyses can be used to report about the robustness of the findings.

Author response: Following the reviewers request we added Faith Index values as the primary alpha diversity measurement, and weighted UniFrac as the primary beta diversity measurement. The Shannon Index plots are saved in **Supplementary Figure 2** as secondary support for the alpha diversity findings. The Bray-Curtis diversity, Aitchison distance and unweighted UniFrac distance are included as additional beta diversity measurements, in **Supplementary Figures 3 and 4**. Main figures (**Figure 2** and **Figure 4**) have been also updated and the following text was added / changed to address this:

(Lines 136-147): "In mothers, SD and PS scores had negative but not significant correlations with α -diversity (within-sample, measured by Faith phylogenetic diversity [PMID:19455206]) based on the bacterial taxonomic composition, consistent with previous reports of decreased α -diversity associated with lower Socioeconomic Status (SES) [PMID:30641975, 26859894, 30249275] (**Fig. 2A**; **Supplementary Table 1F** shows complete correlation statistics). In contrast, among children, α -diversity was significantly positively correlated with SD ($r=0.579$, $n=121$, $P=3.6\times 10^{-12}$) and PS ($r=0.327$, $n=121$, $P=2.5\times 10^{-4}$; **Fig. 2B**). This observation may be partly explained by the significantly lower frequency of breastfeeding among high-SD mothers compared to the low-SD mothers, as breastfed infants have lower GM α -diversity compared to formula-fed infants ($P=3.0\times 10^{-5}$, **Table 1, Fig. 2B**) [PMID:32978424]. Similar results were seen with α -diversity calculated using the Shannon diversity index (**Supplementary Fig. 2**), indicating a robust association regardless of the metric used."

(Lines 148-152): "An examination of dissimilarity (β -diversity, weighted UniFrac distance) between mother and child GMs showed significant negative correlations with SD ($r=-0.438$, $P=5.0\times 10^{-7}$; **Fig. 2C**) and PS ($r=-0.295$, $P=1.0\times 10^{-3}$). This trend holds with several other measures of β -diversity including Bray-Curtis distance, Aitchison distance [Aitchison et al, 2000], and unweighted UniFrac [PMID:20827291] distance (**Supplementary Fig. 3**)."

(Lines 175-194): "We next compared β -diversity between each sample dyad within and between SD and PS groups using weighted UniFrac distance [PMID:20827291] (**Fig. 4**). High-SD mothers had significantly more variable GMs than those of low-SD mothers (FDR-corrected Mann-Whitney U test $P=4.3\times 10^{-6}$; **Supplementary Table 1E**). This effect was even stronger for the PS comparison ($P=2.3\times 10^{-7}$), with low-PS mothers having the most similar GMs out of the comparisons (**Fig. 4A**). High-SD children had the most similar GM profiles (**Fig. 4B**), which potentially relates to their increased α -diversity (**Fig. 2B**). Low-SD children also have some overall similarity in their GM profiles, but the low-SD and high-SD children GMs have little similarity ($P<10^{-15}$ compared to the similarity among low-SD and among high-SD children), consistent with clustering results in **Fig. 3C**. The same is true for PS in the children, except the within-group similarity for low-PS and high-PS was not significantly different. In addition to weighted UniFrac distance (**Supplementary Fig. 4A**), we also identified similar results using other metrics of β -diversity including Bray-Curtis diversity (**Supplementary Fig. 4B**), Aitchison distance (which showed similar trends but less significant results; **Supplementary Fig. 4C**) and unweighted UniFrac distance (which showed more diversity between high-PS samples in the children, and lower distance for high-SD mothers; **Supplementary Fig. 4D**), also indicating a robust statistical association regardless of the metric used. Overall, the greater similarity between GM of low-SD and low-PS mothers compared to high-SD and high-PS mothers suggest distinct environmental exposures that either converge or diverge maternal GM, highlighting the interface of SD/PS, environment and GM."

(Lines 669-673): "For the 16S rRNA/ASV sample analysis, Faith phylogenetic diversity values [PMID:19455206] (for α -diversity) and UniFrac [PMID:20827291] weighted and unweighted distance, as well as Aitchison distance [Aitchison et al, 2000] (for β -diversity) were calculated for each sample using QIIME [PMID:235207646]. Also for β -diversity, Bray-Curtis distance diversity values were calculated using the "vegdist" function in the "vegan" library (version 2.6-4) in R."

R1-5: P24 | 525. "Multivariate testing considering significant covariates..." I think there are two important points to be addressed in this section:.

a) As I will explain, using statistical significance to select covariates is not a valid approach. If one attempts to understand the relation between SD/PS and the gut microbiota, choosing covariates based on statistical significance can bias results. P-values were not designed for this purpose and say nothing about the nature of the relations between variables. For example, a collider (see Wysocki et al. 2022 or Cinelli et al. 2020) would be statistically correlated to predictor and outcome and would induce collider bias if included. A mediator also would be correlated to predictor and outcome and would bias the total effect estimate. Lastly, a variable may be related to predictor and outcome in the expected direction but not pass the threshold of significance; this does not mean it cannot be an important confounder (blocker).

To select covariates, it is advisable to propose a theoretical model, which in turn guides the analysis (see Wysocki et al., 2022). For this reason, I would recommend the authors make clear assumptions like they for example make in a later sentence in this paragraph: "GA for this cohort was restricted to term (≥ 37 weeks), with resulting GA and birthweight differences without precedent for GM differences in the range of values. Maternal age similarly does not have precedence within these age distributions for GM differences beyond that controlled for in SD". Once the authors have described their theoretical model, e.g., in the form of a directed acyclic graph (Wysocki et al. 2022; Cinelli et al. 2020), they can evaluate if their data are compatible with their model(s). Alternatively, the SEM framework allows evaluating several candidate models based on model fit criteria.

Note that if the authors' goal is not to understand the relations but solely to make good predictions, then covariate selection based on statistical significance is still not a good approach. To maximize predictive accuracy, model selection based on information criteria or out of sample prediction performance (e.g., using cross validation) would make sense then. Alternatively, regularized models with broad covariate selection can also make sense for prediction accuracy.

b) The authors write "...not performed on covariates within the SEM model...". Looking at the SEM model in the cited publication, it appears that the covariates the authors seem to refer to are either only reflected in variation in SD or PS. Furthermore, those variables will only load partially onto the latent variables. The remaining variance of each predictor would be ignored. So, the assumption that "correcting for it would eliminate the goal of the analysis." seems questionable to me, especially since the authors perform separate analyses for SD and PS. Nevertheless, I appreciate that the authors state this and several other assumptions and the critical reader is made aware of it. If they could somehow explain/derive that this last assumption can actually be made, that would make the ground of the analyses appear much stronger. Now, it seems shaky to me.

Author response: We appreciate the reviewers' insights into the importance of covariates. In the original manuscript describing the SEM model (previously cited as a preprint, but now published Luby et al. 2023, PMID: 36914799) variables were chosen *a priori* and validated for their hypothesized contribution to the latent constructs SD and PS. For ease, the relevant text from that study is restated here:

Original Luby et. al. text: "We hypothesized that we could dissociate, and find significant independent relationships of, Maternal Social Disadvantage versus Maternal Psychosocial Stress to gestational age and birthweight. To test this hypothesis, we conducted confirmatory factor analyses for **a priori hypotheses** (depicted in Figure 1) about which indicators loaded on these two factors using the MPlus software to validate our grouping of prenatal adversity variables into a Social Disadvantage latent factor (I/N, ADI, Insurance Status, Maternal Education, and Maternal 12 240 Nutrition) and a Psychosocial Stress factor (depression, perceived stress, discrimination, and 241 lifetime measures of trauma and life events)."

The focus of our investigation was to identify bacterial taxa that are good predictors of high or low SD and PS, rather than to understand the underlying relationships with other specific covariates. The PS and SD latent

constructs serve as a useful means of classifying individuals based on a range of variables, and have been subsequently used to identify significant differences in brain composition at birth (PMID:35412624 and PMID:36219693). We have provided the statistical comparisons for the various metadata from the 'high vs low' cohorts (Table 1, Supp Table 1A) for the sake of transparency and to aid the reader in understanding the various factors that may differentiate the groups, but it is not intended as a means of considering how the SD or PS latent factors should be calculated. Identification of distinct and specific predictor taxa for PS and SD provides the evidence for further investigation in examining the specific taxa that are drivers for this difference, including use of animal models as we have in the past (PMID: 27279225).

As a result, and as stated in the text, we did not perform multivariate analysis for the individual components of the latent variables, but rather separated groups for the Random Forest testing based on SD and PS scores. Variables examined not included in the SEM model, that have previously been shown to be related to the infant GM, are included but not controlled for in our Random Forest predictions. We have made revisions to address this throughout the manuscript, including stating this more directly in the second sentence of the results/discussion:

(Lines 112-115): "The calculation of the SD and PS latent factor variables were calculated based on an *a priori* hypothesis (as previously published [PMID:36914799], and previously used to identify significant SD and PS-associated differences in brain composition at birth [PMID:35412624, PMID:36219693])."

In order to reduce confusion on this matter, components of the SD and PS calculation which are not important components for describing the cohort including Perceived Stress Scale (PSS), Healthy eating index, and Income:needs ratio have been removed from the metadata comparison presented in **Supplementary Table S1A**. Other metadata that may be relevant for understanding the cohort such as age and gender are still included.

R1-6: P24 | 543. Significant differences in SD and PS between clusters were tested. Did the authors evaluate absolute measures of cluster strength such as Prediction Strength and Silhouette Index? If those aren't high enough (e.g., ≥ 0.9 and 0.5 , respectively), it is unlikely that these clusters will be reproducible also in other data (see Koren et al. 2013). I'm assuming that the authors are interested in ensuring reproducibility of clusters beyond their data set. But if this is not the case, then the reasons for this should be explained in the manuscript. If, as expected, they would like to ensure reproducibility, the paper by Koren can provide good guidance.

Author response: We have now substantially revised the clustering analysis. Rather than hierarchical clustering, samples are now visualized using NMDS plots. To ensure reproducibility of our clustering we utilized Dirichlet Multinomial Mixtures (DMM) clustering (as suggested by reviewer 2, comment 34) to define the clusters to be visualized on the NMDS plot (**Fig. 3**). We generated statistical support for optimal cluster numbers using DMM log-posterior loss correction values (lplc), Silhouette scores and Prediction strength (as shown in **Supplementary Table 1G**). Taken together, we believe that this improved clustering approach is a much more statistically robust way to present the data. The significant difference in SD and PS in the children according to clustering remained, as seen in the original submission of our manuscript.

The following text was added to the results section and a new figure (**Fig. 3**) was included:

(Lines 153-163): "To further investigate differences between the overall microbiome profiles, we performed the Dirichlet Multinomial Mixtures (DMM) clustering approach for metagenomics [PMID:22319561] to cluster all of the mother and child samples based on the relative abundance of the taxa in their GMs. Using this approach, we identified that the GMs of mothers overall clustered separately from children, with the samples separating optimally into two clusters that separated the children (cluster 1, 94.5% of cluster members) from the mothers (cluster 2, 99.1% of cluster members) (visualized on an NDMS plot in **Fig. 3A**). For all clustering, statistical support for cluster numbers using DMM log-posterior loss correction [PMID:22319561] (lplc), silhouette scores [Rosseeuw et al, 1987] and prediction strength [Tubshirani et al, 2005] is provided in **Supplementary Table 1G**."

(Lines 818-825): **Fig. 3.** NMDS clustering based on 16S relative abundance of taxa in the GM, showing major cluster separations according to Dirichlet Multinomial Mixtures (DMM) clustering. **(A)** Mothers and children group into two clusters, almost entirely separating children (cluster 1, 94.5% of cluster members) from mothers (cluster 2, 99.1% of cluster members). **(B)** Mother samples (N = 121) grouped into two clusters but showed no significant differences in SD or PS. **(C)** Children samples (N = 121) also grouped into two clusters. Cluster 1 has significantly higher SD and PS scores. **(D)** The proportion of feedings with breast milk is shown on the NMDS clustering.

R1-7: P25 | 560. The description of the Random Forest algorithm is insufficient. How did the authors make sure that overfitting was avoided? The descriptions says “generalization error of the model was evaluated by out of bag (OOB) error”. The OOB error is a valid measure of accuracy. However, the authors describe that they selected features before calculating final accuracy. Did they split the data into training and testing data or did they use cross validation to do so? If so, what are the specifics of that ($p = ? / k = ?$). If the authors ran a model on the full data to calculate the importance scores and then reran the model again on the same data only using the most important variables (as described in the paper they cited), this will cause overfitting of the data and all the presented RF accuracy scores are an overestimation of the true accuracy. If this is what the authors did, please report the accuracy of the model that is not overfit. Also, please indicate whether you changed the hyperparameters. It is important to avoid overfitting when tuning models as well. Always make sure that the data you use to evaluate accuracy (here OOB/ROC), has never been “seen” by the model, neither to select features, nor to tune parameters. This is a very common mistake in our field. You may report OOB of the model that has not been tuned (no hyperparameter tuning or feature selection) or you may report accuracy after any kind of tuning, using a test data set that has not been used for tuning/feature selection.

Author response: For details regarding changes made to address the overfitting of the approach please see response R1-2. The updated OOB approach uses only data that has never been “seen” by the model. In the

future, when we generate larger sample sets, we will aim to split the data into training and testing data. However, given the complexity of the human gut microbiome and the number of samples that we have available in the comparison groups, at this time, removing enough training samples to generate an appropriate number of test samples would result in not enough samples to perform a statistically meaningful comparison.

R1-8: P 10 | 211. Are the authors sure that this is how MDA was calculated? Judging from the R packages they used it is more likely that the genome/pathway was not removed but permuted. In case permutation was indeed used, please modify the description.

Author response: We have revised the text to correct the description of the MDA:

(Lines 248-253): "The top 25 strongest predictors of "high" vs. "low" SD (**Fig. 6A**) and PS (**Fig. 6B**) scores in the mothers were identified according to "mean decrease of accuracy" (MDA) scores for each genome (the average decrease in accuracy by randomly permutating the feature values in OOB samples), with additional confidence of differential abundance provided by LEfSe and ANCOM-BC2 analysis shown for each taxa."

R1-9: p 26 | 583. The authors used LEfSe for differential abundance testing. A recent Nature Communications benchmark study (Nearing et al. 2022) was careful to make any clear recommendations but noted that the field may want to avoid LEfSe. Despite this benchmark study, I do not want to ask the authors to entirely replace their LEfSe analysis if they have valid arguments to choose this method over other, in my opinion better, methods. I do however ask the authors in any case to add at least one of the CoDa methods such as ANCOM (II/BC) or ALDEX2 (again, see Gloor et al., 2017 for an explanation of CoDa methods). The results of this additional method can be added to the existing table that shows RF and LEfSe results or otherwise it should be noted somewhere that the results were (or were not) robust depending on the methods used. The reason I ask this is that the field has not yet established a state of the art method that performs best in most scenarios. Therefore, and in order to avoid a reproducibility crisis, we must ensure that findings are robust, independent of method choice.

Author response: We have kept the LEfSe values, but on request of the reviewer we have added ANCOM-BC2 (the most up-to-date version) significance values to **Fig. 6** and **7**, and to **Supplementary Table 11**. The significance values are also described for individual genomes mentioned throughout the text.

MINOR ISSUES

R1-10: Table 1: refers to 'patients'. Should be participants. Also: please explain all abbreviations (i.e. MGS, Psych) in the notes of this and other tables.

Author response: 'Patients' was replaced with 'Participants' throughout the MS. Abbreviations were defined in all Table notes.

R1-11: Fig.1. Legend: clarify what WMS is.

Author response: "Whole metagenome shotgun" has now been defined in the figure and also in the legend.

R1-12: P6 | 134. I find this explanation confusing. Alpha diversity between two samples can be exactly the same and yet the samples can be very dissimilar. The text reads as if alpha diversity is responsible for the result of beta-diversity, while these statistics just reflect different parts of the same snapshot and cannot cause each other. A possible alternative interpretation would be that the high-SD children's GM is more adult-like compared to low-SD children as reflected in higher similarity to their mother's GM and higher alpha diversity.

Author response: We have revised this to read:

(Lines 142-145): "This observation may be partly explained by the significantly lower frequency of breastfeeding among high-SD mothers compared to the low-SD mothers, as breastfed infants have lower GM α -diversity compared to formula-fed infants ($P=3.0 \times 10^{-5}$, **Table 1**, **Fig. 2B**) [PMID:32978424]."

R1-13: p 7 | 146. What led the authors to state that low-SD GMs are presumably healthier? How would they quantify a healthy GM and how did they investigate that the low-SD GMs are more similar to that healthy GM than they have in mind than high-SD GMs? Without any quantification, this statement is not evidence-based and can be misleading. While it may be tempting to point towards what might seem obvious, it is still necessary to do that based on data before making such an interpretation. In the Conclusion section, the interpretation of this finding is much more valid.

Author response: We have removed "who presumably had more consistent, healthy GMs" from this sentence in the results.

R1-14: p 15 | 340. "...although greater SD is associated with greater maternal IL-6 in the full eLABE cohort." Please add a reference for this.

Author Response: This has been removed from the discussion of IL-6, because another manuscript that includes inflammatory markers in the full eLABE cohort is currently under review.

R1-15: p 16 | line 351. "...the impact of exposure to both SD and PS on GM structure and function for mothers and their infants." The study is observational in nature; therefore, causal language (i.e. impact) should be avoided when interpreting results.

Author response: We have revised this to say "the distinct association of exposure to..." instead of "the distinct impact of exposure to..." (Line 99)

R1-16: p 27 | 598. Please consider making your code publicly available such that reviewers can ensure correctness, as well as for other researchers to learn and reproduce your analysis.

Author response: For details regarding this request please see response E-5.

Reviewer #2 (Remarks to the Author):

This reviewer wished to congratulate the authors on an exceptionally well written manuscript. In particular, the methods section and description of statistical analyses are extremely detailed and wholly transparent. The inclusion of substantial supplementary material detailing all test results and associated sample metadata is commendable. Links with SRA accession numbers will facilitate replication findings with ease.

R2-1: The authors refer to plasticity of GM and potential to intervene as an important route to impact for this study. Is the microbiota the driving force behind DS/PS though? Or are environmental factors/exposures associated with DS/PS responsible for shaping GM? Will modulation of GM yield social & psychological equity? This feels like a reach?

Author response: We appreciate and thank the reviewer for bringing appropriate context to the role of the gut microbiome in psychosocial disparities, and not overcalling its role. We have clarified this section to ensure that the GM is not proposed to be a means of modulating psychosocial equity. Discussion is limited to highlighting the interest in the GM due to its potential for alterations. These alterations could potentially diminish any additional contributions the GM makes to health outcomes disproportionately present in disadvantaged populations (i.e. cardiovascular disease/HTN). We have therefore changed the ending paragraph of the Introduction from:

“Understanding the dynamic interaction among the GM and social determinants of health holds promise for interventions because of the potential alterability and plasticity of microbial populations over time, if they are demonstrated to be part of a causal pathway in systemic response to stress.”

To:

(Lines 94-96): “Understanding the dynamic interaction between the GM and social determinants of health is of interest because of the potential for alterability that may lessen potential GM-related health impacts [PMID:26338727, 35382166].”

R2-2: The finding that SD/PS is negatively associated with breastfeeding – which in turn is linked with GM structure/diversity is one of great interest/importance and may inform future practice to understand barriers to breastfeeding in high SD/PS mother-infant dyads as well as promoting breastfeeding within these populations. The authors show that there may be age-dependent dysbiotic mechanisms implicated in high DS/PS states. Predictive features of DS/PS are contrasting between mothers and infants. They illustrate that in infants this dysbiosis may be associated with fewer gut anaerobes such as *Bifidobacterium infantis*, *Veillonella parvula* and *Colinsella* spp. These results tell a lovely story because they fit well with the findings of reduced breast milk feeding in high SD/PS infants and were a joy to read. I would like to see more made of the metabolic pathways though. **If the authors could link metabolic pathways with these strict anaerobes** which appear to be predictive of low SD/PS that would represent a step toward identifying ways in which they might achieve the promises alluded to in the introduction: namely identifying ways to modulate the GM to alleviate impacts of high SD/PS.

Author response: We have emphasized the links to breast milk throughout the text (see responses R2-17 and R2-29). Also, in response to the suggestion in comment R2-26 (below), we have added Figure 8, which shows networks connecting the enriched genomes identified by the analysis with enriched pathways, among the high-SD mother and child results. As described in the associated text, we report on identified links between the anaerobic taxa and the enriched pathways, based on the abundance of the pathway contributed by each taxa.

(Lines 355-369): “The three pathways with the highest predictive value for high-SD in mothers (**Table 2**) related to carbohydrate degradation (sucrose degradation IV, glycogen degradation I, starch degradation III), which appears to relate to the accompanying abundance of strict anaerobic *Bifidobacterium* species that are rich in carbohydrate metabolism pathways [PMID:21484167], with *B. bifidum* and *B. infantis* contributing 62.8% of the total abundance of sucrose degradation IV and 56.5% of the total abundance to glycogen degradation I (**Fig. 8A**). These taxa also contribute to the enriched pathways UDP-N-acetyl-D-glucosamine biosynthesis I and Superpathway of L-threonine biosynthesis, further highlighting their importance not only to the overall SD-defining taxonomic profile, but also to the metabolic potential of the GM. The “myo-, chiro- and scyllo-inositol

degradation" pathway (PWY-7237) was among the most strongly associated with high-PS in mothers (MDA=2.57%, ES=1.5, Kruskal-Wallis FDR-corrected $P=7.4\times 10^{-3}$). Myo-inositol and chiro-inositol degradation by the GM contributes to inositol deficiency [PMID:32670820], which includes metabolic disorders involved with insulin function [PMID:32670820], and MDD when myo-inositol is deficient in the prefrontal cortex [PMID:15953489]."

(Lines 381-387): "The facultative anaerobe *Klebsiella pneumonia* (the fourth-ranked taxa associated with high SD in the children; **Fig. 7A**) was responsible for (i) 30% of the total abundance of the top pathway (pyrimidine deoxyribonucleotides biosynthesis from CTP) and (ii) 33% of the total abundance of the fourth-ranked pathway, superpathway of glycerol degradation to 1,3-propanediol, a function first described in and well-studied in *Klebsiella pneumonia* [PMID:29056473], although the function of this pathway in the gut microbiome is unclear (**Fig. 8B**)."

R2-3: Could the authors validate their predictive factors identified by RF by employing other classification methods such as logistic regression models using only the discriminative features identified?

Author response: In addition to the LEfSe differential abundance comparisons already provided, we have now also added ANCOM-BC2 differential abundance comparisons to provide additional statistical confidence in results. These results are provided on Figures 6 and 7, and Supplementary Table 11 (see also response R1-9).

R2-4: Abstract

Line 62 – “compare cohort” should read “compare cohorts”

Author response: This has been fixed.

R2-5: Line 65-66 – Can the authors provide summary values to quantify these statements? How log is inter-individual GM similarity in high-DS/high-PS mothers? How does this value compare to low-DS/PS mothers? Is this difference significant?

Author response: The corresponding P values for these differences (from Fig 3A) are now indicated in the abstract:

(Lines 66-69): "The lowest inter-individual GM similarity was observed among high-SD/high-PS mothers, suggesting distinct environmental exposures driving a diverging GM assembly compared to low-SD/low-PS healthy controls ($P=3.5\times 10^{-5}$ for SD, $P=2.7\times 10^{-15}$ for PS)."

Due to the abstract word limits, we are not including the average values or measure of similarity here.

R2-6: Line 67 – What is the predictive accuracy of IL-6?

Author response: We have substantially changed the RF approach; we now use a one-round approach instead of the previous two-round approach. The revised text for IL-6 is in "Bacterial species discrimination based on maternal circulating cytokines", with the predictive accuracy of 66.7% (FDR-corrected $P = 0.029$, only significant in the children) is stated in the "Bacterial species discrimination based on maternal circulating cytokines" section of the Results/Discussion (Line 390).

R2-7: Introduction

A very well written and thoughtful introduction to the subject matter. Kudos to the authors!

Line 88 – “interaciton” should be interaction?

Author response: Thank you, we have fixed this typo.

R2-8: Line 89-90 – Could the authors cite some literature describing the importance of gut microbial plasticity and it's therapeutic potential? Suggest something like Marchesi et al., 2016 (<https://doi.org/10.1136/gutjnl-2015-309990>)?

Author response: Thank you for the suggestion, we have added that reference, as well as PMID:35382166.

R2-9: Results

The authors present a vast amount of data in a clear and concise way - congratulations again. The subheadings could be more impactful though. Instead of describing the comparison made, could these highlight the most important findings of those comparisons?

Author response: Thank you, we have edited subheadings to summarize most important findings thus being more impactful.

R2-10: No stratification is explained for the RF models. How many/which samples were used as the training and test datasets? This is extremely important to understand the validity of the model.

Author response: As indicated in the initial submission on line 730 and the Figure 5 caption (lines 839-840), the RF model was evaluated using "out of bag" (OOB) error. In this approach, sample A is removed out of the "bag" of samples, and the rest of the samples "in the bag" are used as training to test whether the sample A is correctly classified. Then sample A is put back into the sample set, sample B is removed "out of the bag", and the training is performed again from scratch with all of the samples in the bag to classify sample B. The process is repeated for all samples, and the accuracy values presented are the result of this approach, in which each sample is classified without its own data being used in the classifier. As indicated by Reviewer 1, (see responses R1-7 and R1-2), this is a valid approach. The two-round approach utilizing the top 25 taxa from the RF was classified as "overfitting" by reviewer 1, therefore the revised version of the paper now only present the RF results using the initial RF OOB accuracy, with additional statistical confidence provided by LEfSe and ANCOM-BC2 results.

R2-11: Multiple figure panels are not referenced in text. This is especially striking for figure 7B. Why are child IL-6 links with microbiota not discussed in text? Especially as the association of multiple Bifidobacterium and Colinsella spp. (which were differential between high/low SD/PS infants) are associated with these stratified groups. It feels as though the results are building up toward something really special but the authors don't really nail selling the big findings at the end??

Author response: We have checked throughout the MS and now all panels are referenced in the main text of the MS.

R2-12: Line-by-line comments included below:

Line 104 – Is the eLABLE study protocol published? Could the authors link out to this here?

Author response: We have now cited the paper by Luby et al, 2023 (PMID: 36914799) as well as the maternal cohort description by Stout et al, 2022 [PMID:36006907] in this sentence (Line 110).

R2-13: Line 105-6 – Assume this means recruitment occurred retrospectively? Could the authors make this clear in both line 102 and the abstract?

Author response: We appreciate the need for clarification regarding study design. Mother –infant dyads for this analysis were drawn from a prospectively enrolled cohort as described in Luby et al 2023 as cited in R2-12. The 121 mother – infant dyads for this analysis were included as prospectively enrolled infants. Plans were made *a priori* to begin GM analysis when all enrolled subjects had delivered, and a majority (90%) of the children reached the first post-natal time point for stool collection, 4 months (355 of 399 delivered infants 89%) to sequence simultaneously and diminish batch effect. All prospectively enrolled infants were included as potential subjects for analysis. To specifically examine the impact of the prenatal environment in this study, we further restricted our analysis to mother who had provided prenatal samples, and eliminated those that were obtained at the time of delivery or after delivery for clarity. Our criteria were established as outlined in **Supplementary Figure 1**.

Plans and rationale for high/low comparisons are outlined in R1-3, outlining the stepwise approach to analysis beginning with 16S rRNA gene sequencing on all samples.

We have revised this line to read:

(Lines 108-110): "A set of 121 mother–child dyads was drawn prospectively from a larger parent study, the Early Life Adversity Biological Embedding and Risk for Developmental Precursors of Mental Disorders Study (eLABLE)", citing Luby et al, 2023 [PMID: 36914799] as well as the maternal cohort description by Stout et al, 2022 [PMID: 36006907].

The abstract has also been revised to read:

(Lines 60-61): "Here, we interrogate the GM of mother-child dyads and maternal circulating cytokines to compare high-versus-low prenatal SD and PS cohorts (**prospective** case-control study design using 16S and shotgun metagenomic sequencing)."

R2-14: Line 106-7 – "SD and PS status had significant variability" is this between high and low groups? This is not made clear in text. Authors' use of the word "significant" implies some sort of statistical testing was performed but no reporting of results. Please specify what the comparison is by which significance is identified or use alternative language such as "substantial".

Author response: Thank you, we have revised this to "substantial" instead of "significant".

R2-15: Line 112 – Appreciate this is a contentious opinion but this reviewer does not consider 16S rRNA gene sequencing to be metagenomics. Consider instead "we performed targeted bacterial 16S rRNA gene sequencing".

Author response: We have revised to this as suggested:

(Lines 122-123) "To provide an overview of GM profiles, **we performed targeted bacterial 16S rRNA gene sequencing** (see Methods) for the 121 mother-child dyads (**Fig. 1**)."

R2-16: Line 130 – Consider adding "significantly" positively correlated with SD and PS?

Author response: We have added this, as suggested:

(Lines 140-142): "In contrast, among children, α -diversity was **significantly** positively correlated with SD ($r=0.579$, $n=121$, $P=3.6\times 10^{-12}$) and PS ($r=0.327$, $n=121$, $P=2.5\times 10^{-4}$; **Fig. 2B**)."

R2-17: Line 132 – Would be useful to reiterate the significance of the difference between high SD/PS mothers and breastfeeding frequency here to really drive this point home. This is a really nice, important finding!!

Author response: We appreciate the reviewers support for highlighting the public health importance of diminished breastfeeding rates identified in high SD/PS. The text has been revised accordingly to highlight these differences:

(Lines 140-145): In contrast, among children, α -diversity was significantly positively correlated with SD ($r=0.579$, $n=121$, $P=3.6\times 10^{-12}$) and PS ($r=0.327$, $n=121$, $P=2.5\times 10^{-4}$; **Fig. 2B**). This observation may be partly explained by the significantly lower frequency of breastfeeding among high-SD mothers compared to the low-SD mothers, as breastfed infants have lower GM α -diversity compared to formula-fed infants ($P=3.0\times 10^{-5}$, **Table 1, Fig. 2B**) [PMID:32978424]."

(Lines 166-173): "We also identified two sample clusters as being optimal for the children GM samples, with cluster 1 having significantly greater SD scores than cluster 2 (Mann-Whitney U test; $P=2.7\times 10^{-9}$), and significantly greater PS scores in cluster 1 than 2 (Mann-Whitney U test $P=6.9\times 10^{-3}$; **Fig. 3C**), suggesting that high-SD and high-PS scores are associated with a distinct overall GM profile in children, but not mothers. As shown in **Figure 3D**, the high-SD / high-diversity GM children tend to have much lower frequencies of breast milk feeding, which has been previously associated with higher GM diversity [PMID:32978424]."

The following text was also added:

(Lines 319-328): "In healthy term infants, low abundance of Bifidobacteriaceae has been linked with systemic inflammation and immune dysregulation early in life [PMID:34143954], development of autoimmunity and early-onset type 1 diabetes [PMID:30356183], as well as neurodevelopment impairment in preterm infants [PMID:34842970]. The microbial differences found in our infants are partly related to variation in breast milk exposure, which is diminished in infants living in high SD/PS (**Fig. 2B** and **2C**). Beyond breast milk, other lifestyle differences have also been shown to have important impacts on early life GM assembly that persist, intergenerationally and well beyond infancy [PMID:35679413]. This highlights the biologic conversion of social disadvantage, in this case linked to breast milk feeding, with the GM and subsequent potential impact on long term health outcomes."

R2-18: Line 139-43 – Another nice finding and fits well to show that SD/PS have a greater impact on GM in earlier than later life. Do the children clustering in the lowest SD/PS scores cluster with their mothers if the authors combined all mothers/children together though? This would work really nicely to illustrate the point that $>SD/PS = GM \text{ diversity among children} = >\text{Similarity to maternal GM}$ and would tie the figure together really nicely.

Author response: To increase reproducibility of our clusters we performed sample NMDS clustering based on the 16S rRNA gene relative abundance of taxa and obtain major cluster separations according to Dirichlet Multinomial Mixtures (DMM) clustering. In addition, the new Figure (**Fig. 3**) now shows that mothers and children also grouped into two clusters, with almost all clustering separating children (cluster 1, 94.5% of cluster members) from mothers (cluster 2, 99.1% of cluster members). Statistical support for cluster numbers using DMM log-posterior loss correction (lpc), Silhouette scores and Prediction strength is shown in **Supplementary Table 1G**.

R2-19: Line 144 – This is confusing. Do you have longitudinal data from participants to compare longitudinal GM profiles? Surely this is the only way to show consistent GM profiles among healthier, or low SD/PS individuals? Is this data not just finding that high SD/PS mothers are more similar to one another. This would suggest some kind of environmental pressure has a homogenising effect on shaping GMs?

Author response: We agree with the reviewer that the wording around having a "consistent, healthy GM" is confusing, and implies we have longitudinal samples within individual subjects for this analysis. This wording has been removed (now Line 176). The point made by the reviewer is exactly the point we hope to highlight, that that high SD/PS mothers are more similar to one another, with environmental influences associated with SD "homogenizing". We have changed the text to better express this point:

(Lines 191-194): "Overall, the greater similarity between GM of low-SD and low-PS mothers compared to high-SD and high-PS suggest distinct environmental exposures that either converge or diverge maternal GM, further highlighting the interface of SD/PS, environment and GM."

In the abstract (line 67) the word "compromised" has been removed.

R2-20: Line 144-57 – B-diversity here is reported as Bray-Curtis similarity!?! Nowhere in methods is a conversion mentioned (assuming $1 - BC\text{-dissim value}$??). If this is the case please state. Otherwise please consider these results may be incorrectly inferred? Is there actually $>\text{dissimilarity}$ between low SD/PS mothers rather than $>\text{similarity}$?? This section may need to be revised but the concept is very interesting. Whether there is a homogenising or individualising influence of SD/PS raises questions about what the environmental factors are that promote these.

Author response: Yes, this was just calculated as "1 - Bray-Curtis dissimilarity", so that the metric for the graphs was similarity, potentially making it easier to interpret. We apologize for failing to define this since it is not common. In the revised version, we use UniFrac weighted distance as the primary beta diversity metric, and Bray-Curtis distance is used in **Supplementary Fig 3B** (See response to reviewer R1-4).

R2-21: Line 168 – Only 11 bacterial genomes frequently detected are listed in text!?

Author response: Thank you for noticing this, we accidentally omitted *B. pseudocatenulatum* from the list but it is now included (Line 205)

R2-21: Line 169-71 – Perhaps reference supplementary figure 3 here?

Author response: Thank you for the suggestion, we now reference Supplementary Figure 3 (new numbering Suppl Figure 6) as indicated:

(Lines 203-207): "Twelve genomes were detected frequently across all samples ($\geq 40\%$ of mothers and children; **Supplementary Fig. 5B**), including five *Bifidobacteria* (*B. breve*, *B. bifidum*, *B. catenulatum*, *B. pseudocatenulatum*, *B. adolescentis*, and *B. infantis*, the most frequently detected genome across all samples; 69.7% of mothers and 89.9% of children; **Supplementary Fig. 6**)"

R2-23: Line 197 – Please reference individual panels of figure 4 that illustrate the points you are making. This comment applies to most of this section of the MS.

Author response: We have revised this particular section as described below, and added more references to figure panels throughout the manuscript:

(Lines 232-235): "First, we used supervised Random Forest (RF [PMID:21039646]) machine-learning to (i) quantify the ability to accurately predict SD and PS classification based on the mothers' and the children's GMs based on binomial distribution tests (**Fig. 5A**; **Supplementary Table 1H**) and receiver operating characteristic (ROC) curves (**Fig. 5B**)..."

R2-24: Line 208-11 – This explanation feels more like methods. Could this text be moved there and the results section here focus more on describing the bacterial features associated with model accuracy?

Author response: Although we acknowledge that this is a long technical description, we feel that it is necessary to highlight the strong significance of the P values, and it is necessary to indicate which statistical test lead to each result here, in order to meet journal requirements for statistics.

R2-25: Line 226 – "Bifidobacteria" should not be italicised.

Author response: This has been fixed.

R2-26: Line 274-6 – Might this suggest age-specific GMs associated with >SD/PS? Are there any pathways that are imbalances between the SD/PS groups across mothers/infants? Could you perform some kind of network analysis to show which pathways are regularly associated with each bacterial genome and how these are implicated in SD/PS grouping?

Author response: In order to explore this, we have now generated some network images (now Figure 8) to visualize the relationship between the genomes and the pathways of high-SD mothers and of high-SD children (the two most interesting groups based on the overall analysis).

The following was added to the text to describe the interesting findings from this analysis:

(Lines 355-369): "Beginning line 353:"The three pathways with the highest predictive value for high-SD in mothers (**Table 2**) related to carbohydrate degradation (sucrose degradation IV, glycogen degradation I, starch degradation III), which appears to relate to the accompanying abundance of strict anaerobic *Bifidobacterium* species that are rich in carbohydrate metabolism pathways [PMID:21484167], with *B. bifidum* and *B. infantis* contributing 62.8% of the total abundance of sucrose degradation IV and 56.5% of the total abundance to glycogen degradation I (**Fig. 8A**). These taxa also contribute to the enriched pathways UDP-N-acetyl-D-glucosamine biosynthesis I and Superpathway of L-threonine biosynthesis, further highlighting their importance not only to the overall SD-defining taxonomic profile, but also to the metabolic potential of the GM. The "myo-,

chiro- and scyllo-inositol degradation" pathway (PWY-7237) was among the most strongly associated with high-PS in mothers (MDA=2.57%, ES=1.5, Kruskal-Wallis FDR-corrected $P=7.4 \times 10^{-3}$). Myo-inositol and chiro-inositol degradation by the GM contributes to inositol deficiency [PMID:32670820], which includes metabolic disorders involved with insulin function [PMID:32670820], and MDD when myo-inositol is deficient in the prefrontal cortex [PMID:15953489]."

(Lines 381-387): "The facultative anaerobe *Klebsiella pneumonia* (the fourth-ranked taxa associated with high SD in the children; **Fig. 7A**) was responsible for (i) 30% of the total abundance of the top pathway (pyrimidine deoxyribonucleotides biosynthesis from CTP) and (ii) 33% of the total abundance of the fourth-ranked pathway, superpathway of glycerol degradation to 1,3-propanediol, a function first described in and well-studied in *Klebsiella pneumonia* [PMID:29056473], although the function of this pathway in the gut microbiome is unclear (**Fig. 8B**)."

Fig. 8. Networks depicting the relative contribution of high-SD associated genomes to high-SD associated pathways in (A) mothers and (B) children. Edges connect high-SD genomes (yellow, from Figures 6 and 7) to high-SD pathways (purple, from Table 2). Node sizes indicate the mean decrease in accuracy (MDA) values from the random forest analysis, and edge thickness and darkness indicate the proportion of reads that each genome contributes to the pathways.

R2-27: Line 322-5 – Again, need to know how many samples used to train / test RF models. Were models built using only cytokines? Could models built using cytokine, functional and taxonomic profiles provide better classification?

Author response: See response R2-10.

R2-28: Line 340 – Seems strange not to reference the full eLABE cohort or provide values to quantify this claim re: IL-6 here?

Author response: See response R1-14.

R2-29: Conclusions

The authors deserve substantial credit for attempting to provide a balanced summary of their data. They explain the important findings but also highlight the limitations of association studies. It is this reviewer's personal opinion that the authors could provide slightly more context to the breast-milk feeding/anaerobic taxa/high IL-6 links in high SD/PS groups though. The authors are almost too cautious with a very exciting dataset.

Author response: We appreciate the reviewer's enthusiasm for our cohort and findings. We have expanded our discussion regarding the impact a relative loss of strict anaerobes early life may have for long term health outcomes and have added the following to our Discussion:

(Lines 485-495): "Our work highlights the previously reported negative relationships between socioeconomic status and breastfeeding rates with a subsequent impact on the neonatal GM [PMID:30247584,25974306]. The resulting GM of highly SD infants at 4 months have greater diversity and a relative paucity of strict anaerobes (*Bifidobacterium*, *Veillonella* and *Collinsella* sp.) reflective of formula feeding. Evidence is accumulating that these early microbial patterns contribute to select immune mediated long term health outcomes over-represented in SD populations including asthma [PMID:27618652], and diabetes [PMID:30356183]. This highlights the biologic conversion of social disadvantage, in this case linked to breast milk feeding, with the GM and subsequent potential impact on long term health outcomes. These results highlight the need for health policy interventions aimed at increasing breast feeding resources for mothers with high SD [PMID:34161260]."

Our examination of maternal IL-6 identified no relationship with maternal GM taxa but that child GM could predict maternal high/low IL-6 with marginal accuracy (66.7% accuracy, FDR-corrected $P = 0.029$). In addition, many of the genomes and pathways significantly associated with low and high IL-6 intersect with those associated with high-SD in the mothers and high-PS in the mothers and children (respectively). We remain cautious in our interpretation of IL-6 results, related to smaller numbers of samples available (N=45-55 per cytokine) clearly identifying this in our cytokine discussion:

(Lines 458-460): "Here, we have identified that many of the genomes and pathways significantly associated with low and high IL-6 intersect with those associated with high-SD in the mothers and high-PS in the mothers and children (respectively)."

We have also added the following to the Conclusions:

(Lines 483-484): "We remain cautious in interpreting these results however, related to limited maternal serum values (n=45-55 samples/group) for these cytokines (methods). "

R2-30: Line 350 – Is this a prospectively assembled cohort? If the selection criteria include reaching 4 months corrected age and are subsampled from the larger eLABE cohort surely recruitment is retrospective??

Author Response: Please see response to R2-13. The prospective nature of the cohort is more clearly outlined in the Methods:

(Lines 527-530): "*A priori* eligibility for this study at four months included if mothers had delivered at ≥ 37 weeks and infants had reached 4 months corrected age (N=355), with both maternal 3rd trimester (T3) and infant 4 months stool samples available (**Supplementary Fig. 1**)."

R2-31: Methods

Line 459 - Please be consistent with capitalisation of Qiime2 throughout

Author response: QIIME2 should be all capitals, and we have revised this throughout except for the mention of the docker container ID which is properly identified in lower-case (qiime2/core:2018.8).

R2-32: Line 475 – Please explain 178 samples used for WMS. 35 low-SD + 43 high-SD + 36 low-PS + 32 high-PS = 146 samples. If only 89 samples from this cohort of 146 were chosen for WMS how were these samples selected?

Author response: The 178 samples represented 89 samples from the mothers and 89 samples from the children. There is some overlap of samples between SD and PS, as indicated in Figure 1C (dark circles). All 89 WMS samples were classified into at least one of high SD, high PS, low SD, or low PS.

R2-33: Line 492 – Was HUMAnN2 or HUMAnN3 used? Tool name is missing – text suggests HUMAnN3 but citation suggests HUMAnN2. Please correct citation if necessary (<https://doi.org/10.7554/eLife.65088>)

Author response: HUMAnN3 was used, and we have updated the reference as indicated.

R2-34: Line 545 – Is hclust the best way to do this? Are 4 clusters per mothers/children the most optimal to describe the communities? Would these numbers be consistent if using DMM and confirming with lowest Laplace approximation score as in Beck et al., 2022, (<https://doi.org/10.1038/s41564-022-01213-w>)?

Author response: We did perform sample clustering using the 16S relative abundance of taxa which showed major cluster separations according to Dirichlet Multinomial Mixtures (DMM) clustering (**Fig. 3**). The statistical support for cluster numbers using DMM log-posterior loss correction (lplc), Silhouette scores and Prediction strength is shown in **Supplementary Table 1G**.

REVIEWERS' COMMENTS

Reviewer #1 (Remarks to the Author):

I appreciate the authors' effort to review the manuscript according to my suggestions. I am very satisfied with the result and have no further points for improvement.

Reviewer #2 (Remarks to the Author):

This reviewer is fully satisfied that the authors have addressed all of my previous comments. The article remains well written and has been strengthened by the inclusion of figure 8 and additional supplementary materials which are now referenced throughout the main text of the manuscript.

The comments below are extremely minor and will take the authors no time to remedy. This reviewer does not need to see any revision once these small edits have been made.

Congratulations to the authors!

Ln 187 – tends = “trends”?

Ln 202 – “identifying 2,219 bacterial genomes across all samples (Supplementary Fig. 5A).” Values in this panel only add up to 1274? (491+404+379)

Ln 204 – “including five Bifidobacteria” = six??

Ln 214-6 – “468 pathways detected across all samples, 94% were in at least 3 mother or 3 child GMs (Supplementary Fig. 5C)”. Values in this panel add up to 438 (9+382+47)

Ln 347 – “ $P=5.2 \times 10^{-7}==$ ” - are these double “==” intended??

Ln 395-6 – “The only circulating cytokine with any significant overall correlation with SD or PS was IL-8 ($P=0.0089$; Supplementary Figure 5)”. Should be Suppl fig 8??

Ln 482 – “there may be alternative pathways contribution to chronic inflammation”. Alternative pathways may contribute to chronic inflammation??.

Reviewer #3 (Remarks to the Author):

I co-reviewed this manuscript with one of the reviewers who provided the listed reports.

REVIEWERS' COMMENTS

1. Reviewer #1 (Remarks to the Author):

1-1 I appreciate the authors' effort to review the manuscript according to my suggestions. I am very satisfied with the result and have no further points for improvement.

Author response: Thank you for your contributions and suggestions in the previous review round.

2. Reviewer #2 (Remarks to the Author):

This reviewer is fully satisfied that the authors have addressed all of my previous comments. The article remains well written and has been strengthened by the inclusion of figure 8 and additional supplementary materials which are now referenced throughout the main text of the manuscript. The comments below are extremely minor and will take the authors no time to remedy. This reviewer does not need to see any revision once these small edits have been made. Congratulations to the authors!

Author response: We thank the reviewer for the detailed suggestions for improvement in the previous draft, and for carefully proofreading the revised version.

2-1 Ln 187 – tends = “trends”?

Author response: Yes, this has been fixed.

2-3 Ln 202 – “identifying 2,219 bacterial genomes across all samples (Supplementary Fig. 5A).” Values in this panel only add up to 1274? (491+404+379)

Author response: In Supplementary Fig. 5a, as indicated in the figure caption, the 1,274 genome counts represent genomes detected in at least 3 maternal samples or at least 3 child samples, to provide confidence of a true detection. This is different than the 2,219 genomes with any detection in any sample. The sentence has been revised to be more descriptive of this difference:

“GM taxonomic profiles were generated from an average of 6 Gb reads/sample using the Unified Human Gastrointestinal Genome (UHGG42) database, identifying 2,219 bacterial genomes across all samples, 1,274 of which were detected in at least 3 maternal samples or 3 child samples (Supplementary Fig. 5A).”

2-4 Ln 204 – “including five Bifidobacteria” = six??

Author response: Yes, this has been fixed.

2-5 Ln 214-6 – “468 pathways detected across all samples, 94% were in at least 3 mother or 3 child GMs (Supplementary Fig. 5C)”. Values in this panel add up to 438 (9+382+47)

Author response: This is the same issue as described in response 2-3. This sentence has been revised to be more descriptive:

“Of 468 pathways detected in any sample, 94% (438) were detected in at least 3 mother or 3 child GMs (Supplementary Fig. 5C).”

2-6 Ln 347 – “P=5.2×10⁻⁷==” - are these double “==” intended??

Author response: This was not intended and has been fixed.

2-7 Ln 395-6 – “The only circulating cytokine with any significant overall correlation with SD or PS was IL-8 (P=0.0089; Supplementary Figure 5)”. Should be Suppl fig 8??

Author response: Yes, this has been fixed.

2-8 Ln 482 – “there may be alternative pathways contribution to chronic inflammation”. Alternative pathways may contribute to chronic inflammation??.

Author response: This sentence has been revised to be more precise:

“The discriminating taxa for IL-6 were distinct from those for SD/PS, indicating there may be different pathways contributing to chronic inflammation than those identified in the SD/PS results (Table 2).”

Reviewer #3 (Remarks to the Author):

3-1 I co-reviewed this manuscript with one of the reviewers who provided the listed reports.

Author response: Thank you for your contributions to the previous review.